

# Snow and glacier melt contributions to streamflow on James Ross Island, Antarctic Peninsula

Ondřej Nedělčev[1], Michael Matějka[2], Kamil Láska[2], Zbyněk Engel[1], Jan Kavan[2, 3] and Michal Jenicek[1]

[1]Department of Physical Geography and Geoecology, Charles University, Prague, 128 00, Czechia
[2]Department of Geography, Masaryk University, Brno, 611 37, Czechia
[3]Alfred Jahn Cold Regions Research Centre, University of Wroclaw, Wroclaw, 50-137, Poland

*Correspondence to*: Ondřej Nedělčev (ondrej.nedelcev@natur.cuni.cz)

**Abstract.** The Antarctic Peninsula is experiencing a rapid increase in air temperature, which has a major impact on the entire
ecosystem, including the runoff process. Although water availability plays an important role in polar ecosystems, runoff
generation in the Antarctic Peninsula region is still poorly understood. We analysed the variability in rain, snow and glacier
contributions to runoff in relation to climate variability in a small, partly glaciated catchment on James Ross Island in the
north-eastern Antarctic Peninsula. We used the hydrological model HBV to simulate the runoff process for the period 2010/11–
2020/21 at a daily resolution. The model was calibrated against both measured discharge and glacier mass balance. Model
simulations showed the negative mass balance of Triangular Glacier for 9 out of 11 study years with an average annual mass
loss of 49 mm water equivalent. About 92% of the annual runoff occurred between October and May. On average, peak runoff
occurred in the second half of the summer season due to the combination of strong glacier and snow melt. The majority (76%)
of runoff originated from snowmelt, 14% originated from glacier melt and 10% from rainfall. The contribution of snowmelt
to total runoff was higher in colder years with more precipitation. In contrast, glacier melt contributed dominantly during
warmer years with less precipitation. Our simulation showed the presence of significant runoff-generating events outside the
usual high summer runoff measurement season.

## 1 Introduction

The Antarctic Peninsula region has experienced a rapid increase in air temperature during the second half of the 20[th] century
(Vaughan et al., 2003; Turner et al., 2005). Although the warming of this region was interrupted in the early part of the 21[st]
century (Turner et al., 2016; Oliva et al., 2017), the region has experienced high air temperatures in recent years (Carrasco et
al., 2021; González-Herrero et al., 2022). Regardless of short-term climate oscillations, future climate projections indicate that
air temperature is expected to rise significantly in both the near and far future (Bozkurt et al., 2021; Zhu et al., 2022).
Increasing air temperature leads to a shift in the phase of precipitation from solid to liquid (Vignon et al., 2021), which
accelerates the snow cover melting (Abram et al., 2013), glacier mass loss (Cook et al., 2005; Vaughan, 2006; Chuter et al.,
2022; Seehaus et al., 2023) and changes in the permafrost active layer (Hrbáček and Uxa, 2020; Kaplan Pastíriková et al.,



2023). These changes have a significant impact on the runoff regime of catchments in proglacial areas (Gooseff and Lyons, 2007; Nowak et al., 2021). Streams in proglacial environments represent a key driver for changes in the entire ecosystem, as they form the landscape through fluvial erosion and sediment transport (Rosa et al., 2014; Kavan et al., 2017). Streamflow variations and water availability have a major influence on the evolution of terrestrial ecosystems (Gooseff *et al.* 2017), and

changes in the amount of freshwater and sediment released into coastal seas can affect marine ecosystems (Meredith et al., 2018; Braeckman et al., 2021).

In Antarctica, runoff is mainly studied in the McMurdo Dry Valleys (Chinn and Mason, 2016; Gooseff et al., 2022). In this area an increase in the flow season duration has been observed (Gooseff and Lyons, 2007). However, climatic conditions in the McMurdo Dry Valleys differ significantly from those in the Antarctic Peninsula region. Annual precipitation in the

McMurdo Dry Valleys is less than 100 mm (Doran et al., 2002; Fountain et al., 2010), several times lower than the modelled amount of 300–700 mm on James Ross Island in the north-eastern Antarctic Peninsula (Van Wessem et al., 2016).

In the Antarctic Peninsula region, runoff has been studied on King George Island, James Ross Island, the South Orkney Islands and Vega Island. Nonetheless, these studies only covered one summer season and focused mainly on sediment transport (Rosa et al., 2014; Hodson et al., 2017; Sziło and Bialik, 2017; Kavan et al., 2017; Stott and Convey, 2021; Kavan, 2021; Kavan et

al., 2023)(Stott and Convey, 2021; Kavan et al., 2017; Kavan, 2021; Hodson et al., 2017; Rosa et al., 2014; Sziło and Bialik, 2017; Kavan et al., 2023). A limited number of studies have focused on the runoff generation process (Moreno et al., 2012; Lyons et al., 2013; Lee et al., 2020). The most complex study in this field so far was conducted by Jung *et al.* (2022), who described the different contributions of old and new water during warm and cold periods on King George Island based on the chemical and stable isotopic water composition. Falk *et al.* (2018) analysed the variability of surface runoff in a small partly

glaciated catchment on King George Island during one summer season using a simple hydrological model developed by the authors. Kavan *et al.* (2017) analysed the runoff variability of two streams on James Ross Island for the 2014/15 summer season. Their results indicate that the runoff variability of these streams was mainly driven by air and ground temperatures, with maximum runoff occurring at the end of January. The subsequent study (Kavan, 2021) focused primarily on fluvial sediment transport, but also described runoff variability for the 2017/18 summer season.

Although changes in the runoff process affect both terrestrial and marine ecosystems, runoff generation in the Antarctic Peninsula region is still poorly understood. This is partly due to the severe limitations of direct measurements of individual components of the precipitation-runoff process, which can only be made during the short austral summer season and are often subject to large measurement errors (Tang et al., 2018; Seefeldt et al., 2021). Therefore, the objectives of our research were 1) to reconstruct streamflow in a small, partly glaciated catchment on James Ross Island, Antarctic Peninsula, and 2) to assess

the inter-annual variations in rain, snow and glacier contributions to streamflow in relation to recent climate variability. The study area belongs to the largest deglaciated area in the Antarctic Peninsula, with ongoing changes in water source contributions to streamflow affecting the seasonal runoff distribution, which further impacts terrestrial ecosystems. To our knowledge, studies on the runoff process in the Antarctic Peninsula region have only been conducted for individual summer seasons. Therefore, the inter-annual variability of the runoff process has not been investigated. To fill this gap, we used a





conceptual hydrological model coupled with a glacier mass balance model to perform runoff simulations of individual water balance components over the 2010/11–2020/21 period based on available climate, glacier mass balance, snow depth and streamflow data.

## 2 Data and methods

### 2.1 Study area

The analyses were performed in the Triangular Glacier catchment, which is situated in the Ulu Peninsula in the northern part of James Ross Island (Fig. 1). The catchment area covers 2.8 km$^2$ and elevation ranges from 70 to 588 m a. s. l. The catchment is situated on the Lachman Crags plateau, its south-western slopes, and adjacent low-elevated plain forming part of Abernethy Flats to the west. The highest part of the catchment spreads over the Lachman Crags plateau (~400 to 600 m a.s.l.), which is separated from the rest of the catchment by up to 300 m high cliffs (Engel et al., 2023). The lower part of the catchment is

covered by glaciers, ice-cored moraines, and braid plains (Jennings et al., 2021). A small, shallow ice-marginal lake is located near the terminus of Triangular Glacier.




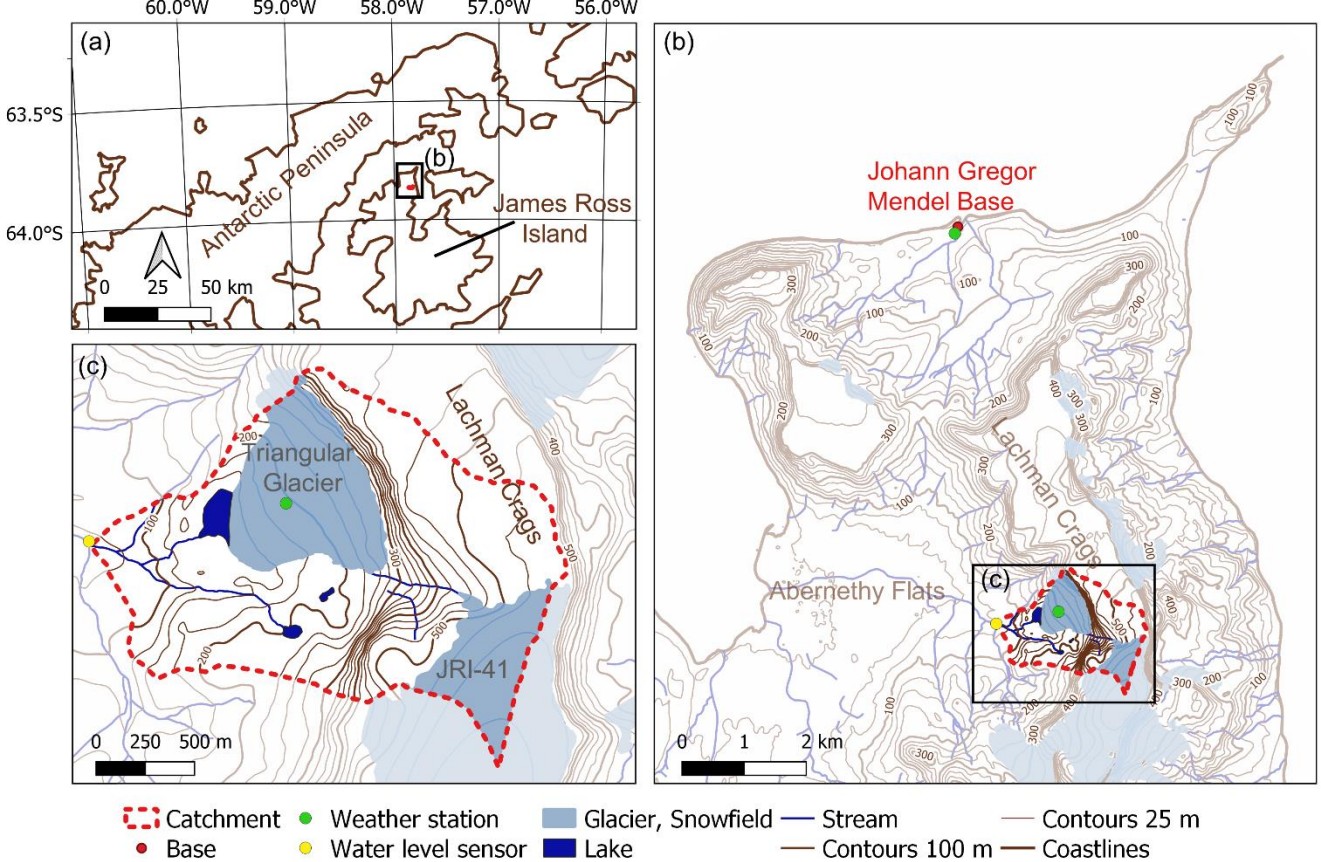

**Figure 1: Location of the study catchment in the northern part of James Ross Island (Matsuoka et al., 2018; Czech Geological Survey, 2009).**

The catchment area was delineated from the digital terrain model produced by GEODIS (Czech Geological Survey, 2009) for the northern part of James Ross Island at 8 m spatial resolution. Almost one third of the catchment (28%) is glaciated, 19% of the catchment is covered by Triangular Glacier (GLIMS Glacier ID: G302151E63856S), which represents the smallest glacier on James Ross Island (Engel et al., 2023). The south-eastern part of the catchment (9%) is covered by the northernmost tip of the Lachman Crags ice cap (GLIMS Glacier ID: G302158E63874S).

The mean annual air temperature measured on the north coast of James Ross Island near the Johann Gregor Mendel Station (Fig. 1b) was -6.7 °C for the period 2011–2021 (Ambrozova et al., 2019; Kaplan Pastíriková et al., 2023). Estimated annual precipitation ranges from 300 to 700 mm of water equivalent (Van Wessem et al., 2016). Snow depth in the lowland plains of James Ross Island is typically less than 30 cm (Hrbáček *et al.* 2016). In contrast, seasonal changes in snow depth varied from 30 to 90 cm in Triangular Glacier (Engel et al., 2023). This uneven accumulation of snow is largely due to the prevailing strong
south-westerly winds, which have a significant impact on both snowfall distribution and snow redistribution by wind



(Kňažková et al., 2020; Kavan et al., 2020). According to Engel et al., (2023), snow drift accounts for 43% reduction in snow depth in Triangular Glacier.

The evolution of Triangular Glacier followed the changes in air temperature. From 1979 to 2006, the glacier lost almost one third of its volume. After that, the rate of retreat decreased for almost a decade. Since 2015, the retreat rate has increased again
(Engel et al., 2023).

## 2.2 Data

The semi-distributed bucket-type hydrological model HBV in its software implementation HBV-light version 4.0.0.25 (Seibert and Vis, 2012; Seibert and Bergström, 2022) was used to simulate the runoff process for the period 1 June 2010 to 31 May 2021. 1 June was estimated as the start of a water year. Mean daily air temperature and total precipitation were used as input
to the model. Daily runoff and seasonal glacier mass-balance series were used for model calibration (see below).

Air temperature was measured at the Johann Gregor Mendel Station (Fig. 1b) using the Minikin TH datalogger and EMS33 temperature sensor (EMS, Brno, Czechia) with an accuracy of ±0.15 °C. A linear regression was used to adjust the time series based on air temperature measurements performed in the middle part of Triangular Glacier during the 2017/18 summer season. The temperature-based method defined by Oudin *et al.* (2005) was used for the calculation of daily potential evapotranspiration
which was one of the inputs to the hydrological model.

Direct measurement of precipitation, mostly occurring as snowfall, is very difficult and inaccurate in the study area due to strong winds, which limits the usage of standard approaches, e.g. heated rain gauge. Therefore, daily precipitation was simulated by the Weather Research and Forecasting model in version 4.3 (WRF; Skamarock *et al.* (2019)). The model was run in a two-nested-domain setup with a horizontal resolution in the inner domain of 5 km. In the vertical coordinate, 65 eta-levels
were used for the simulation of atmospheric dynamics. The WRF model was forced by ERA5 reanalysis (Hersbach et al., 2020). The WRF model has been successfully applied in previous studies on James Ross Island (Matějka et al., 2021; Matějka and Láska, 2022). Following these validation studies, the model used the 3DTKE boundary layer scheme (Zhang et al., 2018), Thompson microphysics scheme (Thompson et al., 2008) and NoahMP (Niu et al., 2011) as a land surface scheme dealing with interactions between the surface and the atmosphere. Both shortwave and longwave radiation were parameterized with
the RRTMG scheme (Iacono et al., 2008).

The time series of runoff for the period 8 February–15 March 2018 was calculated from automatic water level measurements (in 10-min interval; station indicated with a yellow dot, Figs 1 (b,c)) using a hydrostatic pressure sensor (DipperLog F100/M30, Heron Instruments, CA) and the rating curve derived from manual velocity measurements (Flowtracker Handheld Acoustic Doppler Velocimeter, SonTek, USA).

The seasonal surface mass balance of Triangular Glacier was estimated by the glaciological method once a year for the years 2014/15–2019/20. The initial volume of Triangular Glacier at the start of model calibration period (2015) was calculated based on the ground penetrating radar survey performed in 2017 (Engel et al., 2023). The volume of Triangular Glacier at the start of the simulation (2009) was derived from ice surface digital elevation model of Triangular Glacier in 2006 (Engel et al.,



2023). To account for the change in glacier extent between 2006 and 2009, we adopted the value of -0.3%/a[-1] reported by
Engel et al., (2023) as the mean annual retreat rate between 2006 and 2014.

Snow depth was measured using a sonic distance sensor (Judd Communication, USA) fixed at the automatic weather station located in the central part of Triangular Glacier during the period 6 February 2017 to 23 January 2020. The snow depth time series was used only for model validation. Global solar radiation was measured using CMP11 pyranometer (Kipp & Zonen, the Netherlands) near the Johann Gregor Mendel Station from 1 January 2015 to 10 March 2018. The global radiation time
series was only used for the correlation analysis between climate and runoff data.

## 2.3 Hydrological model HBV coupled with glacier routine

For runoff simulations, the HBV model was used (Seibert and Vis, 2012). The model consists of five routines, 1) a degree-day method-based snow routine, which controls snow accumulation and melt, including snow water holding capacity and potential refreezing of meltwater, 2) a degree-day method-based glacier routine, which simulates glacier mass balance, 3) a soil moisture
routine, which simulates actual evapotranspiration and groundwater recharge, 4) a groundwater routine in which runoff from two groundwater boxes is calculated, and 5) a routing routine which simulates propagation of runoff through the catchment. The more detailed model description is given by Seibert and Vis (2012).

The catchment was divided into sub-zones based on elevation intervals of 100 m for glacier-free areas, and 50 m for glaciated areas. The ice water equivalent (WE) was calculated using the Δh parametrization method that links glacier area and thickness.
This method is implemented in the model and it is described in detail in Seibert *et al.* (2018). The warming-up period was one year. The effect-tracking algorithm implemented in the model and described in Weiler *et al.* (2018) was used to determine the contribution of rain, snowmelt, and glacier melt water to runoff with the assumption of complete mixing of water in individual model components (Jenicek and Ledvinka, 2020). The snow redistribution function was set up in the model, which redistributes snow above a certain amount from elevations above a defined threshold to lower elevations. Specifically for the study
catchment, snow accumulations above 1000 mm which occurred above 525 m a. s. l. were redistributed evenly over the lower zones. A small section of the Lachman Crags ice cap located in the highest part of the catchment was not considered in the model.

The model was calibrated against both runoff (available only for the 2017/18 summer season) and glacier mass balance using a genetic algorithm procedure (Seibert, 2000) for the period 2017/18–2019/20 with the warming-up period 2014/15–2017/18.
Two objective functions were used for calibration; 1) Kling-Gupta efficiency KGE (Gupta *et al.* 2009; 70% weight) which combines the evaluation of the bias of mean runoff, flow variability and flow dynamics, and 2) mean absolute error of glacier water equivalent (30% weight). To reduce the issue of parameter equifinality, a median simulation which was calculated from one hundred calibration runs was used for further analysis. The model performance was additionally evaluated using an approximately three-year long time series of recorded snow depth.





## 2.4 Runoff and climate characteristics and data analysis

Several runoff and climate characteristics were used in the analysis. Mean daily or annual air temperature ($T$), precipitation ($P$) and snow water equivalent ($SWE$), annual sum of positive air temperature ($T_{positive}$) and average daily global radiation solar (GR) were used to describe climate variability (Table 1). All of the above characteristics represent mean catchment values. Additionally, glacier water equivalent ($WE$) was analysed separately for each elevation zone. Runoff variability was represented by daily, monthly and annual runoff ($Q$) and by rainfall, snowmelt and glacier melt runoff components ($Q_{rain}$, $Q_{snow}$ and $Q_{glacier}$ respectively). Spearman's correlation coefficient ($r_s$) was calculated to analyse the relationship between runoff and climate characteristics.

| Characteristic | Description |
|---|---|
| $Q$ | Runoff [mm] |
| $Q_{rain}$ | Rainfall contribution to runoff [mm] |
| $Q_{snow}$ | Snowmelt contribution to runoff [mm] |
| $Q_{glacier}$ | Glacier melt contribution to runoff [mm] |
| $T$ | Air temperature [°C] |
| $T_{positive}$ | Sum of positive air temperature [°C] |
| $P$ | Precipitation [mm] |
| $GR$ | Global solar radiation [W m$^{-2}$] |
| $SWE$ | Snow water equivalent [mm] |
| $WE$ | Glacier water equivalent [mm] |

**Table 1: Runoff and climate characteristics used in the analyses.**

## 3 Results

### 3.1 Evaluation of the model performance

The values of the objective functions resulting from the model calibration showed good performance for both runoff and glacier mass balance (Fig. S1). The median values of 100 calibration runs were 0.33 and -11.9 for Kling-Gupta efficiency and glacier absolute mean relative error, respectively. For selected objective functions not used for calibration, the median values were 0.32 for Nash-Sutcliffe efficiency, 0.64 for runoff volume error, 0.83 for runoff Spearman rank correlation coefficient and 0.98 for glacier water equivalent, absolute mean relative error.

A comparison of observed and simulated runoff showed that the model simulated the daily variability of runoff relatively well, although peak runoffs were generally underestimated (Fig. 2).



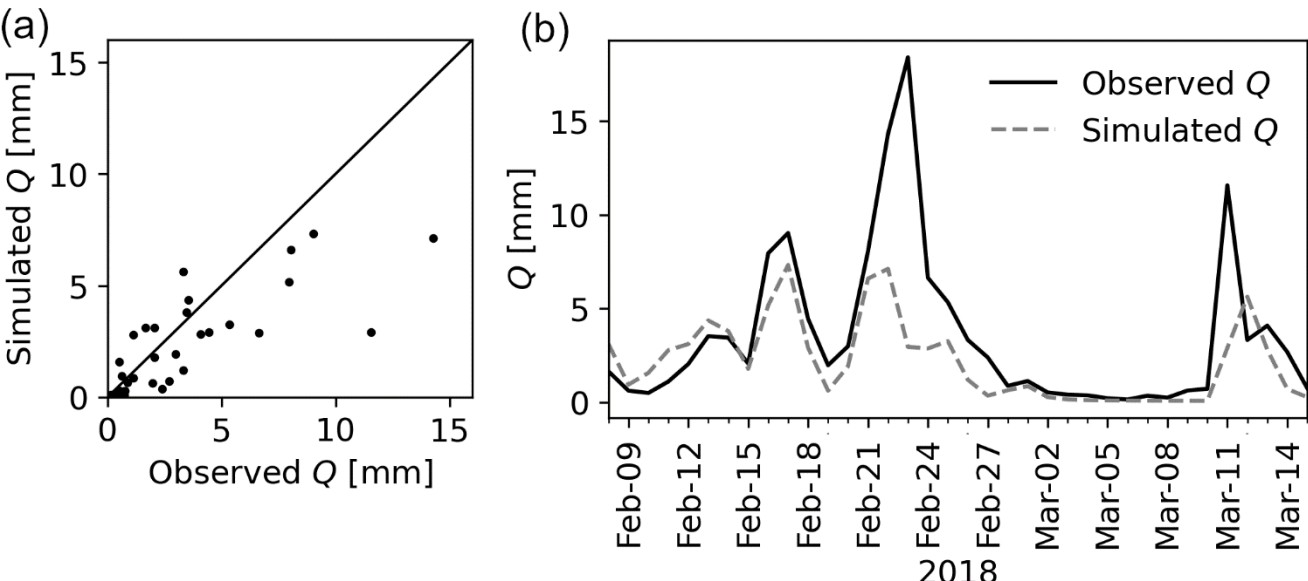

**Figure. 2: Simulated versus observed runoff ($Q$) in Triangular catchment for the period 8 February–15 March 2018.**

The simulated glacier mass balance matched well with the observed glacier mass balance for years with available measurements (cumulative difference 7.6 mm of *WE* for the period 2014/15–2019/20; Fig. A2a). For all years but one (2015/16), the simulated mass balance was within the estimated uncertainty of the measured mass balance (Fig. A2a).

The model was further evaluated by comparison of simulated snow water equivalent and the observed snow depth. There was a strong correlation between observed and simulated values, although the absolute values differ (Pearson correlation coefficient equal to 0.82; Fig. A2b). The fact that the observed snow depth was not used for model calibration suggests the model's ability to simulate the snow storage sufficiently well.

**3.2 Variability in snow cover and glacier mass balance**

The model results showed that the average annual mass loss of Triangular Glacier was -49.7 mm (± 1 mm considering the variability of 50% of the model simulations given by different calibrated parameter sets). The negative mass balance during the first two water years of the study period was followed by the highest positive mass balance (19 mm) in 2012/13, followed by two years of near-equilibrium mass balance in 2013/14 and 2014/15 (Fig. 3a). From 2015/16 until the end of the study period, the mass balance remained negative. The highest mass loss occurred in the last year of the study period in 2020/21 (-150 mm). The second highest mass loss was found in 2016/17 (-130 mm).





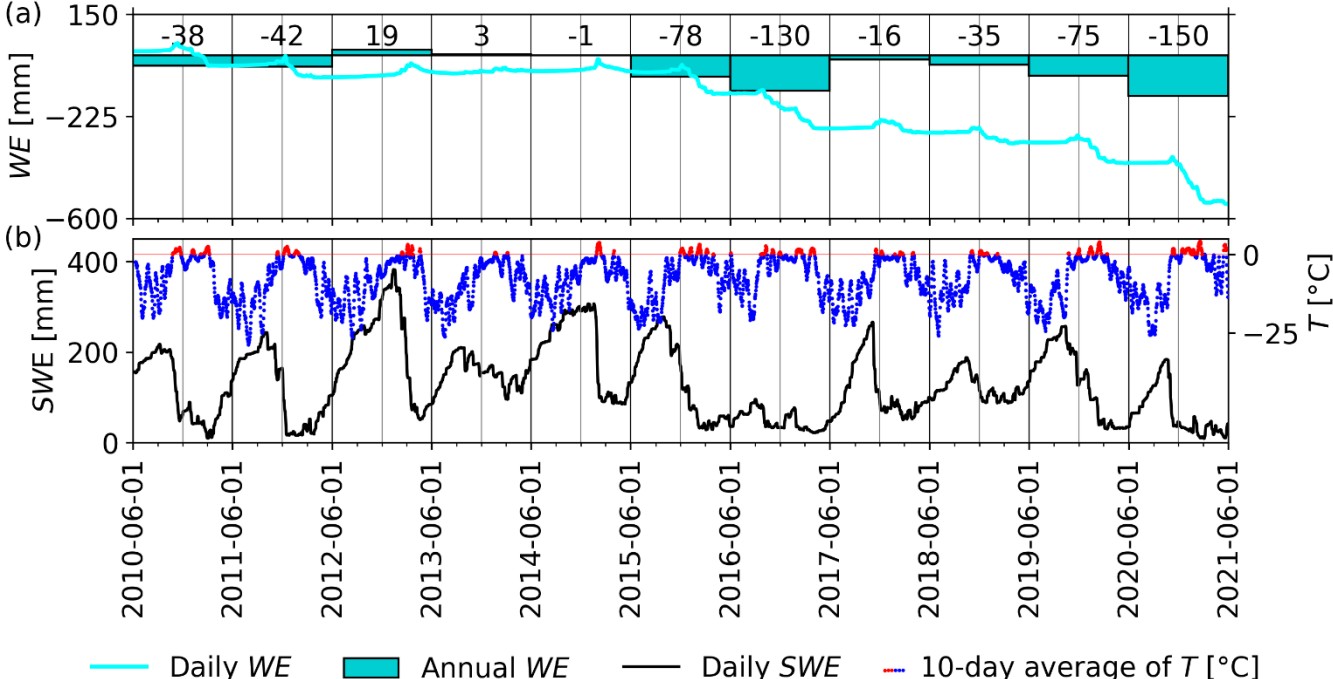

**Figure 3: (a) Cumulative daily changes in glacier water equivalent (WE) from the start of the simulation (cyan line), and changes in WE in individual water years (bars). (b) Daily snow water equivalent SWE (black line) and 10-day moving average of air temperature (negative values in blue, positive values in red).**

The lower middle part of the glacier, at elevations between 150 and 200 m a.s.l., the second largest part of the glacier by area, experienced the largest absolute decrease in glacier water equivalent, with a reduction of 217 mm. In contrast, the highest parts of the glacier did not melt at all.

Mean catchment daily *SWE* ranged from 59 mm on 18 March to 216 mm on 29 September. The highest *SWE* (382 mm) occurred on 12 January 2013 (Fig. 3b). It was caused by an unusually cold beginning of summer, resulting in almost no snowmelt until late January (see Fig. 6a). The lowest annual maximum *SWE* was 9 mm, which occurred on 4 March 2021, associated with very high air temperatures during the warm part of the year (Fig. 3b). Monthly *SWE* loss was highest in November and February. These months accounted for 32% and 25% of the total *SWE* loss, respectively.

### 3.3 Runoff regime

The simulated mean annual runoff for the study period was 415 mm (± 45 mm). The majority of runoff (92%) occurred from October to May. Mean annual evapotranspiration was 7 mm and mean annual precipitation was 369 mm (Fig. 4). About 76% (315 ± 39 mm) of the runoff originated from snowmelt, 14% (58 ± 5 mm) from glacier melt and 10% (42 ± 7 mm) from rainfall. The flow regime is characterised by two peaks, one in November and the second in February. During the winter, air temperature was well below the freezing point and thus almost no runoff occurred. In general, as the air temperature rises at the start of the warm season and causes snow to melt, the contribution of $Q_{snow}$ to the total runoff increases. In contrast, $Q_{glacier}$



210 starts to contribute during summer, as parts of the glacier are often snow-free. $Q_{rain}$ contributes to total runoff throughout the warm season, primarily from October to March (Fig. 4).



**Figure 4: Mean daily rainfall runoff ($Q_{rain}$), snowmelt runoff ($Q_{snow}$), glacier melt runoff ($Q_{glacier}$) (areas, left axis), mean daily precipitation (P; blue bars), 3-day moving average of air temperature (T; red line), cumulative Q, $Q_{rain}$, $Q_{snow}$, and $Q_{glacier}$ (lines, right**
215 **axis) in the Triangular catchment for the period 2010/11−2020/21**

In Figs 4 and 5, we can see that throughout the year, $Q_{snow}$ was the primary contributor to runoff. From December to March, the mean monthly $Q_{glacier}$ exceeded the mean monthly $Q_{rain}$.

Several distinct peaks can be seen in the mean monthly values: $Q_{snow}$ in November and February; $Q_{rain}$ in November and January; $Q_{glacier}$ in February (Fig. 5).





**Figure 5: Mean (bars), minimal and maximal (error bars) monthly runoff ($Q$), rainfall runoff ($Q_{rain}$), snowmelt runoff ($Q_{snow}$) and glacier melt runoff ($Q_{glac}$) in the Triangular catchment for the period 2010/11–2020/21.**

The mean monthly runoff was highest in February, accounting for 18.3% of the annual runoff (Fig. 5). The high runoff during this month was primarily caused by the combination of the highest monthly $Q_{glacier}$ and the second-highest monthly $Q_{snow}$. In November, which had the highest mean monthly $Q_{snow}$, the second-highest mean monthly runoff (17% of the annual runoff) was recorded. Additionally, November also had the highest mean monthly $Q_{rain}$. In contrast, the lowest monthly runoff occurred in August accounting for only 0.3% of the annual runoff. The runoff was also very low in June and July (0.9% and 0.7% of the annual runoff).





### 3.4 Inter-annual runoff variability

Annual runoff varied between 282 and 501 mm during the period of 2010/11–2020/21 (Table 2). In 2013/14, the lowest annual runoff occurred due to a cold summer (as can be seen in Fig 3) resulting in the lowest annual $Q_{rain}$ (17 mm), low annual $Q_{snow}$ (260 mm), and almost no annual $Q_{glacier}$ (4mm). The highest annual runoff was observed in 2019/20 as a result of the second-highest annual $Q_{snow}$ (375 mm) combined with above-average $Q_{rain}$ (45 mm) and above-average $Q_{glacier}$ (79 mm).

|  | Annual runoff [mm] | | | | Contribution [%] | | |
|---|---|---|---|---|---|---|---|
|  | Q | $Q_{rain}$ | $Q_{snow}$ | $Q_{glacier}$ | $Q_{rain}$ | $Q_{snow}$ | $Q_{glacier}$ |
| 2010/11 | 442 | **100** | 280 | 59 | **23** | 63 | 13 |
| 2011/12 | 412 | 53 | 306 | 51 | 13 | 74 | 12 |
| 2012/13 | 414 | 27 | **382** | **3** | 7 | 92 | **1** |
| 2013/14 | **282** | **17** | 260 | 4 | 6 | **92** | 1 |
| 2014/15 | 388 | 18 | 352 | 18 | **5** | 91 | 5 |
| 2015/16 | 477 | 22 | 370 | 85 | 5 | 78 | 18 |
| 2016/17 | 412 | 47 | **237** | 128 | 11 | **58** | 31 |
| 2017/8 | 413 | 46 | 343 | 24 | 11 | 83 | 6 |
| 2018/9 | 342 | 44 | 256 | 42 | 13 | 75 | 12 |
| 2019/20 | **501** | 45 | 375 | 79 | 9 | 75 | 16 |
| 2020/21 | 489 | 39 | 299 | **151** | 8 | 61 | **31** |
| Mean | 415 | 42 | 315 | 58 | 10 | 76 | 14 |

**Table 2: Annual Q, $Q_{rain}$, $Q_{snow}$ and $Q_{glacier}$ and contribution of $Q_{rain}$, $Q_{snow}$ and $Q_{glacier}$ to runoff.**

The contributions of rainfall, snowmelt, and glacier melt to runoff varied significantly from year to year. The lowest annual $Q_{glacier}$ occurred in the year 2012/13 (3 mm; Table 2), and the contribution to runoff was 1% (Fig. 6a). Over this year, the highest annual $Q_{snow}$ (382 mm) was observed. This was due to the cold beginning and middle of the summer, resulting in the largest snow accumulation during the observed period, which did not melt until the end of the summer (Fig. 3b). In both years 2012/13 and 2013/14, $Q_{snow}$ contributed the most to the total runoff, accounting for 92% of it.



(a)

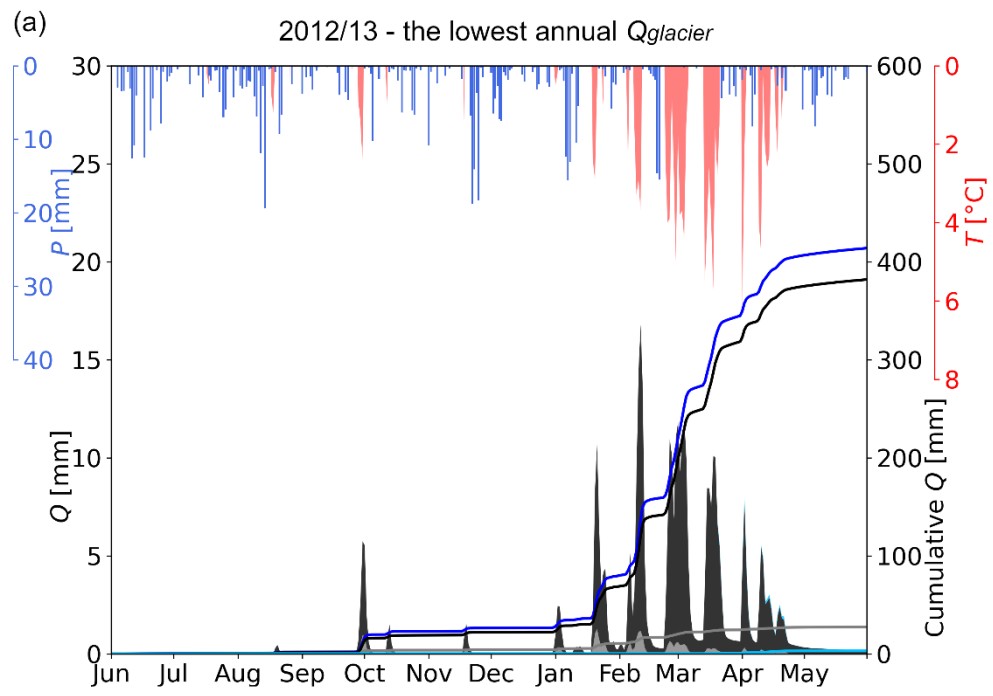

(b)

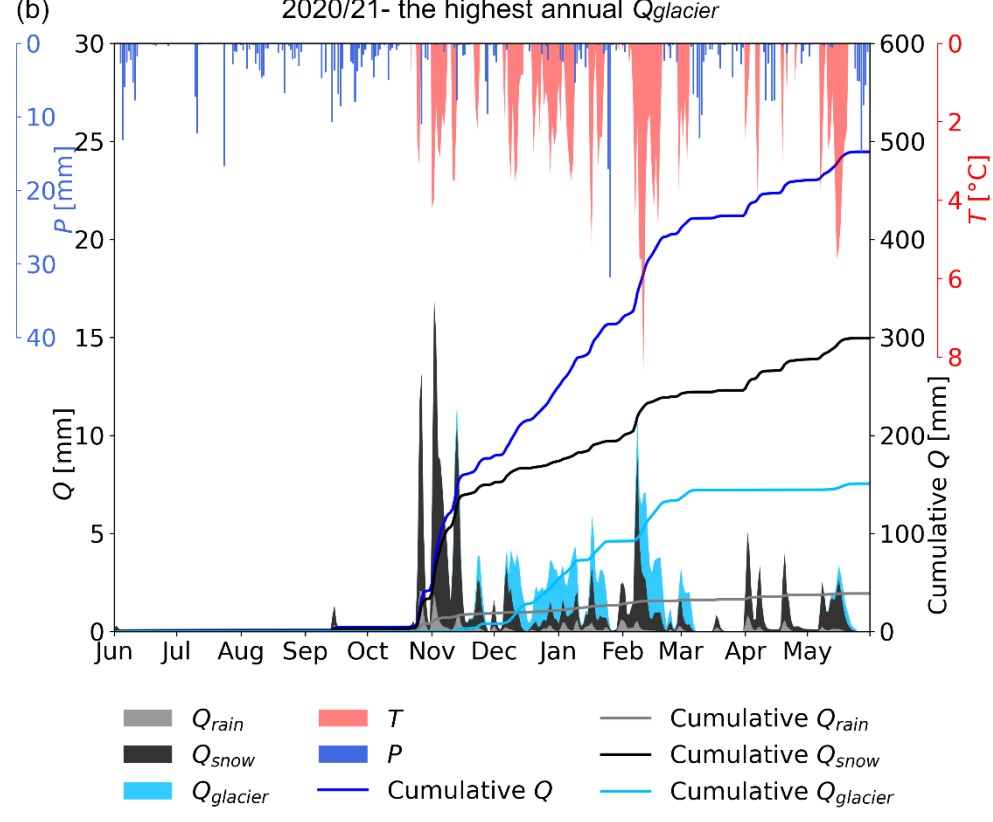





**Figure 6: Contribution of rainfall runoff ($Q_{rain}$), snowmelt runoff ($Q_{snow}$), glacier melt runoff ($Q_{glacier}$) to total runoff ($Q$) during two selected years; (a) the year 2012/13 with the lowest annual glacier melt contribution and (b) the year 2020/21 with the highest glacier melt contribution.**

The highest annual $Q_{glacier}$ (151 mm) occurred in 2020/21 (Table 2, Fig. 6b). This was due to the large number of days with high air temperature from November to May (72 days with positive daily air temperature from November to February), which caused the snow cover to melt-out earlier than in other years (Fig. 3b). In 2020/21, $Q_{glacier}$ accounted for 31% of the annual runoff. Despite below-average annual $Q_{rain}$ and $Q_{snow}$, the annual runoff was the second-highest (489 mm) in 2020/21. In 2016/17, annual $Q_{glacier}$ also contributed 31% to total runoff. This was due to the low snow accumulation during the winter

(Fig. 3b) which caused the lowest annual $Q_{snow}$ (237 mm).

In 2010/11, the highest annual $Q_{rain}$ occurred (100 mm; Table 2), which accounted for 23% of the total annual runoff. This is an unusually high amount, as the second-highest annual $Q_{rain}$ was only about half of the previous year's amount (53 mm in 2011/12).

**3.5 Seasonality in peak flows**

In addition to examining the inter-annual variability of runoff, we analysed individual extreme runoff events and their causes. Most of the annual peak flows occurred from November to the first half of February. Only in 2013/14 did the annual peak occur at the start of October (Fig. 7). These annual peak flows were caused by different water sources contributing to the total runoff at different times. Annual peak flows were consistently associated with high $Q_{snow}$ and occurred during the same runoff

episode as annual peak $Q_{snow}$, except in 2015/16. In that year, the annual peak runoff in January was due to a combination of annual peak $Q_{glacier}$ and high $Q_{snow}$, while the annual peak $Q_{snow}$ occurred in October.





**Figure 7: Daily rainfall runoff ($Q_{rain}$), snowmelt runoff ($Q_{snow}$), glacier melt runoff ($Q_{glacier}$) (lines) and maximal values of Q, $Q_{rain}$, $Q_{snow}$, $Q_{glacier}$ (triangles) for individual water years.**

The annual peak $Q_{rain}$ typically occurs during October and November in most years. However, in 2012/13, 2014/15, and

2018/19, the annual peak $Q_{rain}$ occurred in January, February, and December, respectively (Fig. 7). The annual $Q_{glacier}$ peak

mostly occurred in February, except for January in 2011/12 and 2015/16, and April and March in 2012/13 and 2013/14,

respectively (Fig. 7). However, in these two years (2012/13 and 2013/14), almost no $Q_{glacier}$ was observed (see Table 1).



### 3.6 Impact of climate on runoff variability

To identify the climatic factors that show the greatest impact on runoff variability, we conducted a correlation analysis. This analyses of climate and runoff characteristics showed strong positive correlations between annual $T_{positive}$ and both annual runoff, $Q$, and annual $Q_{glacier}$ ($r_s = 0.68$ and $0.82$, respectively; p-values $< 0.05$; Fig. 8a). A weaker correlation was found between annual $Q_{glacier}$ and annual $T$ ($r_s = 0.63$; p-values $< 0.05$). Additionally, annual $Q_{glacier}$ showed a negative correlation with annual $P$ ($r_s = -0.62$, p-value $< 0.05$). In contrast neither annual $Q_{rain}$ nor annual $Q_{snow}$ showed a significant correlation

with annual $T$ or annual P. Among the three runoff components, only annual $Q_{glacier}$ showed a significant correlation with total annual $Q$ ($r_s = 0.59$ p-value $< 0.1$), indicating that $Q_{glacier}$ has dominant control over the variability of total annual runoff.

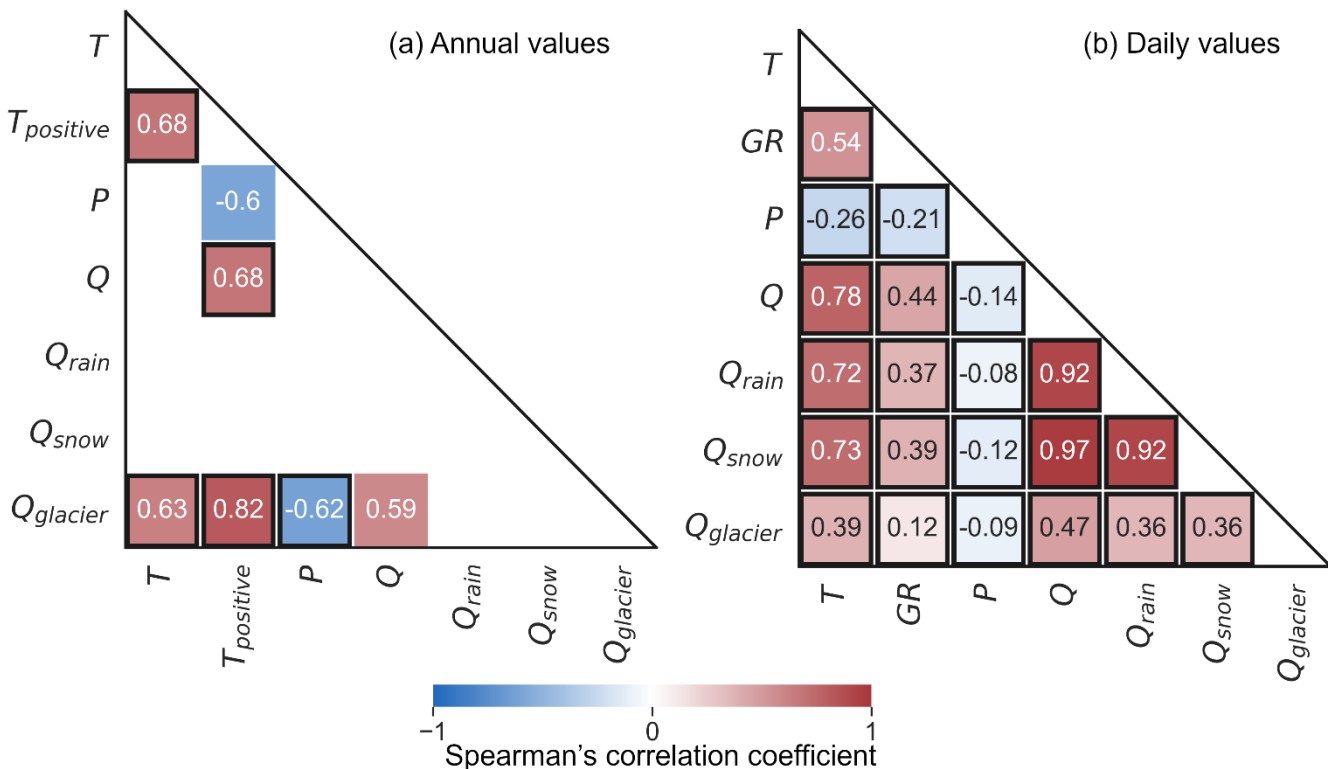

**Figure 8: Spearman's rank correlation coefficients for the selected meteorological and hydrological characteristics at (a) annual and**
**(b) daily level. The colours indicate the value of the Spearman's rank correlation coefficient. Correlations with p-value < 0.1 are shown with a coloured rectangle without a border, correlations with p-value < 0.05 have a black border.**

The correlation analysis for daily values was conducted on a subset of the data (from 1st January 2015 to 10 March 2018) for which measured global radiation was available. This analysis showed that daily $Q$, $Q_{rain}$, $Q_{snow}$ and $Q_{glacier}$ were more strongly correlated with daily air temperature ($r_s = 0.78$, $0.72$, $0.73$, $0.39$, respectively, p-values $< 0.05$) than with daily global radiation

($r_s = 0.44$, $0.37$, $0.39$, $0.12$, respectively, p-values $< 0.05$; Fig. 8b) A significant, although weak, negative correlation was found between daily runoff characteristics and daily precipitation ($r_s > -0.15$, p-value $< 0.05$). All daily runoff components were



strongly correlated with daily runoff. However, correlations involving $Q_{glacier}$ were much weaker than those associated with the other runoff components.

The highest annual runoff occurred during warmer (positive anomaly of $T_{positive}$) and drier years (negative anomaly of precipitation; Fig. 9a). Such years rather occurred in the second half of the study period (solid circle margins in Fig. 9). The high annual runoff in these years was associated with a higher relative contribution of the glacier to runoff. (Fig. 9d). All four years with above-average $Q$ are among the five years with above-average $Q_{glacier}$. Conversely, in years where $T_{positive}$ was below average and precipitation was above average, the relative contribution of snowmelt to runoff was higher. (Fig. 9c). No specific pattern was found for $Q_{rain}$ (Fig. 9b).

Figure 9: Relationship between absolute anomaly of $T_{positive}$, relative anomaly of $P$ and relative anomaly of (a) mean annual runoff Q, (b) rainfall runoff $Q_{rain}$, (c) snowmelt runoff $Q_{snow}$ and (d) glacier melt runoff $Q_{glacier}$ for the years 2010/11–2014/15 (dashed circles) and 2015/16–2020/21 (solid circles).





## 4 Discussion

### 4.1 Input data and hydrological modelling uncertainty

The modelling of the precipitation-runoff process is subject to uncertainties, which must be considered in the interpretation of the results. Uncertainties are mostly associated with the model structure and with the amount and accuracy of input and calibration data. Our study catchment is located in a very remote, poorly accessible region, where limited data are available for model calibration. For example, streamflow data are typically only available during the austral summer when the polar

station is in operation (Kavan et al., 2017; Kavan, 2021).

Additionally, the streamflow measurements themselves are more uncertain than in more accessible regions because it is not possible to construct a standard gauge profile. Therefore, the measured profile (cross-section) may change during the measurement period which additionally causes lower accuracy of the respective rating curve. Due to this constraint, the HBV model was chosen for the water balance components simulations as it has been found, despite several limitations, to be suitable

for simulating the runoff process in polar environments (Wawrzyniak et al., 2017; Osuch et al., 2019, 2022). Additionally, it has been repeatedly tested in catchments where only very short or episodic streamflow time series are available, or even single measurements exist (e.g. Seibert and McDonnell 2015, Pool *et al.* 2017). The HBV model structure is rather simple compared to physically based models, but it has been shown to achieve better results than more complex models (Seibert and Bergström, 2022; Girons Lopez et al., 2020). However, a limitation of the HBV for modelling runoff in polar regions is that it doesn't

include soil thermal processes (Bui et al., 2020).

Another possible source of uncertainty is the WRF-modelled precipitation. However, when compared to observations in summer 2021/2022, this model performed very well (Matějka et al., 2022). The model was validated using four nested domains in resolutions 8100 m, 2700 m, 900 m and 300 m. For all domains, the bias for cumulative precipitation from 2 January 2022 to 26 February 2022 reached -16 % to -1 % with the Spearman´s rank correlation coefficient for daily amounts of 0.86 to 0.87.

This suggests a relatively low sensitivity of simulated precipitation accuracy to model resolution in contrast to, e.g., wind speed in complex terrain (Matějka and Láska, 2022). The impact of the WRF model resolution on simulated precipitation in the Antarctic Peninsula region was discussed also by Pishniak and Beznoshchenko (2020).

The above data limitations may be partially overcome by calibrating the model against multiple components of the runoff process, leading to better overall performance (Konz and Seibert, 2010; Finger et al., 2015; Nedelcev and Jenicek, 2021). In

the case of our study catchment, the HBV model was calibrated against both runoff and glacier mass balance. In addition, simulated *SWE* correlated well with measured snow depth and *SWE* (Fig. A2b). All of the above procedures reduced model uncertainty and increase overall model performance and reliability (van Tiel et al., 2020).

Another source of uncertainty in our study is not considering the Lachman Crags ice cap in the hydrological modelling, although a part of the glacier is located in the highest part of the study catchment at elevations above 500 m a.s.l. Unfortunately,

there are no measurements of the mass balance of this glacier. Since we did not consider this glacier in the hydrological model, only snow accumulation and snowmelt were simulated from this part of the catchment. This could lead to an underestimation



of $Q_{glacier}$. However, we assume the overall influence of the glacier on the total catchment water balance as negligible since our model did not simulate any glacier melt above an elevation of 350 m a.s.l. (Fig. 3a). Additionally, the glacier is covered with snow most of the year preventing any potential glacier melt.

## 4.2 Mass balance of Triangular Glacier

In their study, Engel et al., (2023) linked the changes in Triangular Glacier mass balance to air temperature variability. The simulated mass balance of Triangular Glacier is generally consistent with the direct mass-balance measurements by Engel et al., (2023), who reported a low melt rate between 2006 and 2014, followed by a period of large mass loss. The simulated mass balance of Triangular Glacier was negative in all water years except for 2012/12 and 2013/14. In the water year 2014/15, the simulated balance was slightly negative (-1 mm; Fig.3). However, if the year would be defined by dates of mass balance measurement, the simulated mass balance would be positive (20.25 mm; Fig. A2a) which is in agreement with (Engel et al., 2023). The difference between the simulated and measured mass balance may be due to the location of the glacier on the leeward side of Lachman Crags, which has a significant effect on snow accumulation (Kavan et al., 2020).

The simulated mass balance variability from 2010/11 to 2013/14 corresponds well with the mass balance pattern of nearby Whisky Glacier and Davis Dome glaciers, as described by Engel et al., (2018). Both of these glaciers had a positive balance in these years, except for 2011/12, which was the year with the highest simulated mass loss for Triangular Glacier from 2010/11 to 2013/14. Similar to Davis Dome, the simulated mass balance of Triangular Glacier was also slightly negative in 2014/14, while Whisky Glacier had a positive balance in that year. However, in contrast to the positive mass balance for Davis Dome and Whisky Glacier, our simulation shows a negative mass balance in 2010/11. This could be attributed to the different definition of the start of each year.

In addition, it is worth mentioning that according to Engel, Láska, Matějka, et al. (2022), the summer of 2021/22 was even warmer than the previous year, which was the last year included in this study and the one with the highest simulated mass loss. (Fig. 3a).

## 4.3 Runoff regime

The results showed that the flow season in the study catchment lasts from September to May, with snowmelt being the main contributor to runoff throughout the year. Based on mean monthly runoff the seasonal hydrograph may be characterized by two peaks (Fig. 5). The first peak, occurring in November, was mainly caused by snowmelt. The second and higher peak, occurring in February, was a result of a combination of snowmelt and glacier melt contributing to the runoff. In most years, the highest daily runoff was observed in January or February, and it was associated with peak $Q_{snow}$.

The occurrence of the maximum runoff at the turn of January and February is in agreement with conclusions by Kavan *et al.* (2017), who measured streamflow in two proglacial streams near the Johan Gregor Mendel Station during the 2014/15 summer season (Fig. 1b). However, these results are limited to measurements of less than two months. Gooseff and Lyons (2007), based on long-term measurements in the McMurdo Dry Valleys, reported that peak flows occur at the beginning of the flow



season. The reason for this finding may be that the total precipitation amount is lower compared to our study area, resulting in
lower snow accumulation on glaciers and thus its earlier melt-out, causing the earlier initiation of glacier melt.

On King George Island, runoff variability is primarily driven by air temperature associated with snowmelt at the beginning of
the season (Falk et al., 2018). This is because the climate is warmer, and the region experiences higher rainfall during the
summer season compared to our study area. Towards the end of the summer season, runoff variability is driven by rain events.
The results showed high inter-annual variability of runoff mainly associated with glacier melt. The persistent mass loss of
small land-terminating glaciers around AP may lead to the depletion of water storage in catchments and larger dependency of
runoff generation on seasonal snow cover will very likely lead to increased variability of runoff (Huss and Hock, 2018). This
in turn may cause water shortages and affect local freshwater ecosystems. Glacier melt also supports sediment delivery to the
ocean providing the marine ecosystems with important minerals and nutrients (Hodson et al., 2017). Suspended sediment
transport is expected to increase in the near future due to rising air temperatures (Stott and Convey, 2021). Of the three
components, rain contributed the least to total runoff, however, an increase in rainfall frequency and intensity is expected in
the future (Vignon et al., 2021). Higher amounts of rainfall can also affect the melting of snow and glaciers.

**4.4 Impact of climate on runoff variability**

Inter-annual variations in runoff were strongly influenced by variations in air temperature, with higher total runoff observed
in warmer and drier years with a higher relative contribution of glacier melt to runoff. In contrast, the relative contribution of
snowmelt to total runoff was higher in years with relatively higher precipitation and lower air temperatures. This was due to
the fact that more snow was accumulated and therefore melted later in the season in these years, leaving the glaciers snow-
covered for most of the summer season which resulted in low glacier melt.

Mean daily runoff showed a stronger correlation with air temperature than with global radiation, which is consistent with the
findings of Kavan et al. (2017) and Falk et al. (2018). Additionally, Kavan et al. (2017) reported a similarly strong relationship
between ground temperature at 5 cm depth and discharge in glacier free catchments, which contrasts with the weak relationship
between ground temperature and discharge in glaciated catchment reported by Falk et al. (2018).

**Conclusions**

We simulated water balance and daily runoff in a small partly glaciated catchment on James Ross Island over the period
2010/11–2020/21. Specifically, we analysed the inter-annual variability of glacier, snow, and rain contributions to total runoff.
The main findings can be concluded in the following points:

The mean annual runoff for the 2010/11–2020/21 study period simulated by the HBV model was 415 mm. Together with mean
annual evapotranspiration (7 mm) and mean annual precipitation (369 mm). The water budget is largely influenced by the
negative mass balance of Triangular Glacier (for 9 of 11 years) with an average annual mass loss of -50 mm WE. About 76%
of runoff originates from snow cover, 14% from glaciers and only 10% from rainfall.





The majority (92%) of annual runoff occurred between October and May, with the highest mean monthly runoff occurring in the second half of summer due to a combination of strong glacier- and snow melt. Additionally, high runoff was found in November due to the rapid melting of the seasonal snow cover. In all months, $Q_{snow}$ was the dominant contributor to runoff. Of the three runoff components, only annual $Q_{glacier}$ was significantly correlated with total annual Q, indicating the dominant control of $Q_{glacier}$ on inter-annual runoff variability. The contribution of snowmelt to total runoff was higher in colder years

with more precipitation (mostly snowfall), while glacier melt contributed more during warmer years with overall less precipitation.

Overall, the results showed that runoff on James Ross Island is highly variable from year to year. The main driver of this variability was the variability in air temperature, which affected the variation in annual glacier runoff more than annual snowmelt runoff. Our simulation showed the presence of runoff-generating events outside the usual summer runoff

measurement season. These warm events contribute significantly to the total annual runoff and are usually omitted in conventional observational studies that only cover the peak melt season.

**Appendix A**

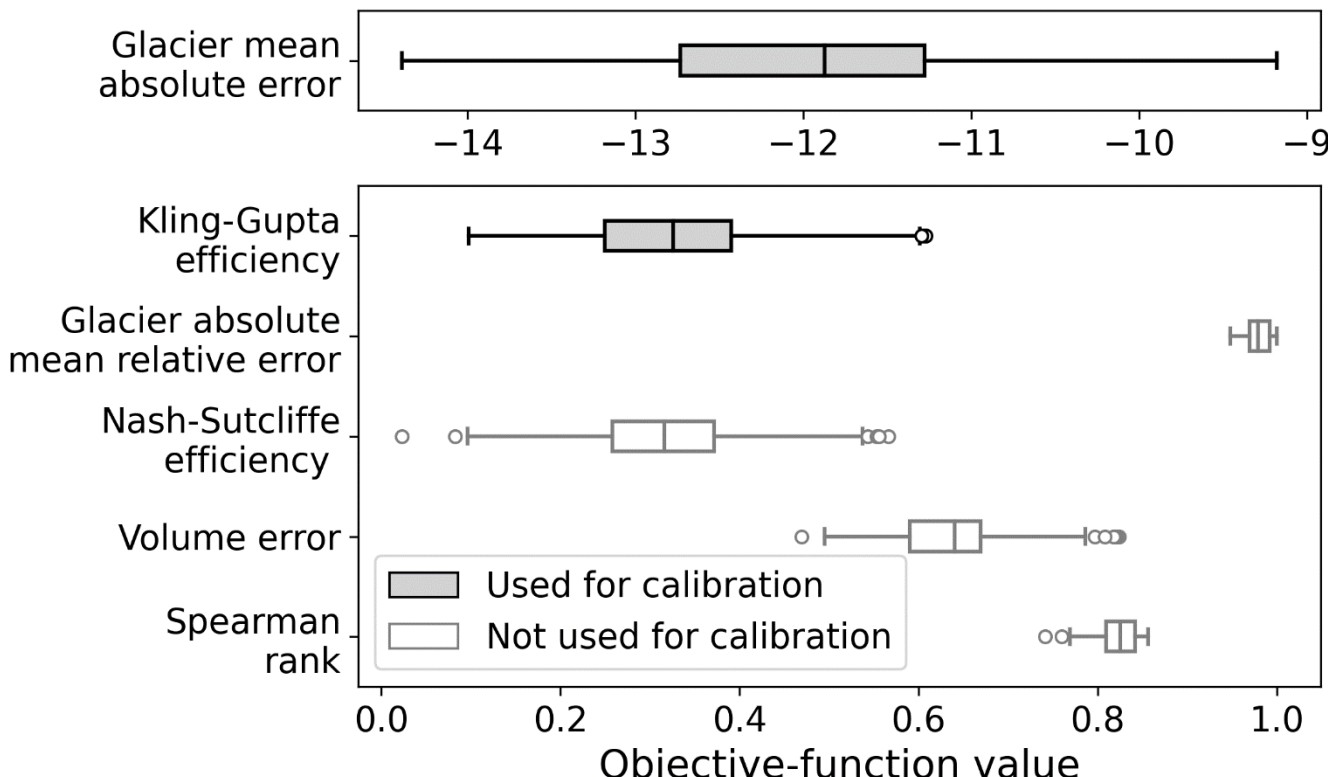

**Figure A1: The objective function values for one hundred calibration runs. Black are functions used for calibration of the model.**




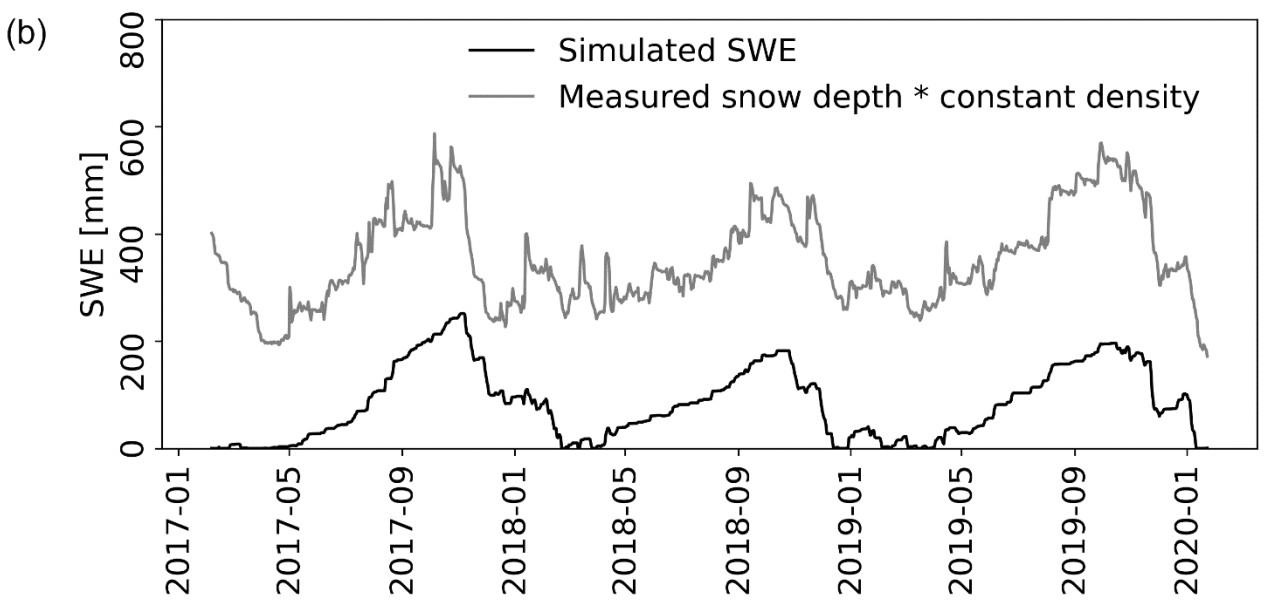



**Figure A2: (a) Simulated and observed mass-balance. Observed mass-balance with estimated uncertainty was taken from (Engel et al., 2023). (b) Mean simulated SWE in the elevation zone, in which the weather station is located and estimated SWE which was calculated as observed snow depth on Triangular Glacier (with zero point level that was chosen arbitrarily (Engel et al., 2023))**
**multiplied by constant density 389 kg·m$^{-3}$ which was the mean measured density in summer season 2022.**

## Data availability

The HBV model outputs was published as an open dataset(Nedělčev et al., 2024).

## Author contribution

MJ and ON designed the study;
ON performed hydrological modelling and analysed the data;

MM performed meteorological modelling;

KL, ZE and JK provided meteorological, glaciological and hydrological data

ON and MJ wrote the manuscript draft;

MM, KL, ZE and JK reviewed and edited the manuscript

## Competing interests

The authors declare that they have no conflict of interest.

## Acknowledgements

Support from the Czech Science Foundation (project no. 20-20240S) and from the Johannes Amos Comenius Programme (P JAC), project No. CZ.02.01.01/00/22_008/0004605 are gratefully acknowledged. This work was additionally supported by the
Ministry of Education, Youth and Sports of the Czech Republic (e-INFRA CZ; ID:90254). Many thanks are due to Jan Seibert and Marc Vis from the University of Zurich for providing valuable consultations and feedback regarding the HBV model calibration and evaluation, and Tracy Ewen for improving the language. We also thank to the crew of Johann Gregor Mendel Station for the extensive fieldwork support and to EMS Brno (Czechia) for the long-term support and advice in meteorological measuring systems.

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
