# Peer review of "Snow and glacier melt contributions to streamflow on James Ross Island, Antarctic Peninsula"

_EGUsphere, 2024_

## Author Comment (AC1)

Thank you for the review of our manuscript. We appreciate your constructive comments and suggestions. Please find our point-by-point response below (in blue).
In general, we have identified the following main issues, which we have addressed below:

1) We found the method section related to a description of the HBV model (structure, routines, parameters) too brief and we will describe it in more detail.

2) We will be more explicit about the data used for model calibration and simulation. In particular, we will clarify how the discharge, glacier mass balance and snow data have been used to model calibration and evaluation

3) We will better explain how the runoff components are calculated in the model and how to interpret the results related to the inter-annual variability of the components and explain the contradiction between the snowmelt which contributes most to the total runoff and glacier-melt which controls the inter-annual variability.

4) We agree that more emphasis should be placed on a literature review and discussion related to understanding the runoff process in polar areas and its impact on terrestrial ecosystems.

Overall, this manuscript is within the scope of the journal and adds some significant contributions to our understanding of the water budget and runoff processes on the Antarctic Peninsula. As I understand it, this paper is about quantifying the various contributions from snow, glacial melt, and rain to streamflow on the Antarctic Peninsula, and the relationships of total runoff and runoff contributions to climate variations. However, the title only mentions snow and glacial contributions but the paper also addresses rain contributions as well. There is also some confusion with the various study periods for the different model components as well as the field validations and measurements. The methods section needs more detail on the model: how the model uses the input data and how it estimates many of the simulated values, and also the field validation methods and what exactly was measured and how it was used. The discussion section also needs to be expanded in order to elaborate on the importance of the results and the implications for the ecosystems in this region and the effects of climate on runoff processes.

**Specific comments**

The title only mentions snow and glacial contributions but the paper also addresses rain contributions as well.

When formulating the research questions, we were interested mostly in snow and glacier contributions since the rainfall is rather minor. However, we agree that the original title might be confusing, so we have decided to change the title accordingly to "The role of snowmelt, glacier melt and rainfall in streamflow dynamics on James Ross Island, Antarctic Peninsula"

1. ABSTRACT AND INTRODUCTION:

Lines 13-14: It would be helpful to actually state what the total study period is (i.e. June 2010 – May 2021).

We agree that this change might improve readability. We will reformulate the respective text to "We used the hydrological model HBV to simulate the runoff process from June 2010 to May 2021 at a daily resolution."

Lines 17: What does "strong glacier and snow melt" mean?

We found the word "strong" to be too vague and redundant in the sentence. We will remove it from the text.

Lines 25: What are "high air temperatures in recent years"? Can this be quantified above average or above previous maximum temperatures?

We agree that the text is too vague. We will rephrase it to "Although the warming of this region was interrupted in the early part of the 21st century (Turner et al., 2016; Oliva et al., 2017), the cooling period ended in the mid-2010s and (Carrasco et al., 2021) since then the region has experienced several warm events with record high temperatures (González-Herrero et al., 2022)."

Line 40: It may be helpful to also note that all of the precipitation in the dry valleys is in the form of snow (whereas the peninsula also receives some rain, analyzed in this paper).

We will add the note to the text: "In addition, this region receives a certain amount of precipitation in the form of rain, unlike the Dry Valleys where only snowfall was observed (Fountain et al., 2010)."

Line 55 and Line 62: It would be helpful to include some description of the terrestrial and marine environments in these regions in relation to streamflow and how they are affected by these runoff processes. What is the importance of understanding runoff processes for these ecosystems and environments?

The description of the impact of changes in terrestrial ecosystems on streamflow is stated in L30-35. However, we will do more literature search on the topic and extend this part accordingly.

Lines 66: Again the study period of 2010/2011 – 2020/2021 is confusing. It may be helpful to state that streamflow only occurs during XX months, so the study covers the time period of June 2010 – May 2021 or something along those lines…

We agree and we will reformulate the text to "… water balance components from June 2010 to May 2021 based on available climate …"

1. METHODS:

Lines 86-87: How much of the precipitation is estimated to come from rainfall and from snow each year? Has this changed over time?

As previously stated on L106, numerical models represent the only available source of information about precipitation from a long-term perspective in the study area. However, data from three stations located west of the Antarctic Peninsula Carrasco and Cordero, (2020) indicate an increase in precipitation from 1970 to the early 1990s, followed by a decrease

from 1991 to 1999. Additionally, an increase in snowfall and a decrease in rainfall were observed from the mid-1990s to the mid-2010s, resulting from the cooling period. We will consider adding this information to the text.

Lines 70 and on… Some further discussion of the streams where discharge is being measured would be helpful. How much does the discharge vary from year to year? What is the average flow season? Is runoff only estimated at one location/stream?

Indeed, in such remote areas, the water level is not possible to measure continuously over longer periods. Further uncertainties arise from discharge calculation using rating curves which need to be created separately for each year due to the riverbed changes. Usually, the discharge data is only available from a few locations around the polar station and only during its operation (usually one to two months each year). For our study area of the Triangular catchment, the only available measurements cover the time period from February to March 2018 which was used as one of the variables for the HBV model calibration (see the respective lines 149-152 of the original manuscript). Therefore, neither the inter-annual variability nor the length of the discharge season are known. However, based on your comment, we decided to extend the discussion on the above topic.

Line 99: Why was 1 June estimated as the start of the water year?

After a careful literature search, we are not aware of any systematically used definition of the hydrological year in this area. Therefore, we determined the beginning of the hydrological year based on our data. We have chosen June as the beginning because (based on our dataset) from that month onwards the number of days with precipitation and air temperature above 0 °C is negligible.

Line 99: Were air temp and total precip the only inputs for the model? How does it use those to estimate runoff and glacier mass balance?

The model inputs are formed by air temperature (T), precipitation (P) and potential evapotranspiration (PET; calculated using air temperature). Available data of glacier mass balance and runoff was used for model calibration. The whole procedure is described in section 2.3. However, we now realized that the model description is probably not fully clear and some important details are missing (e.g. the better description of snowmelt and glacier routines), although references to detailed model descriptions are provided in the original text. We have decided to substantially extend the methods section to better explain how the model deals with input data and how the individual water balance components are calculated.

Lines 106: Was there any field validation for the simulated precipitation?

Yes, there was a field validation of the simulated precipitation. We will add the following text to the respective section: "Short-term precipitation measurements were taken in January and February 2022 at the Johann Gregor Mendel Station using a Thies laser precipitation monitor (disdrometer) and a manual rain gauge. These data were used to validate the performance of the Weather Research and Forecasting (WRF) model (Matějka et al., 2022)."

Lines 116: Why was the runoff only calculated from Feb – Mar 2018? Was there no other data outside this period? How did this period relate to the rest of the study period?

Please also see our response to one of your similar comments above. There was no other measured discharge data in the study catchment outside this period. We are aware that this period represents only a small part of the entire study period and this was one of the reasons why we used glacier mass balance measurements for model calibration. As noted above we will extend the discussion section related to the topic with a new paragraph (also related to your comment on lines 304-305).

Lines 120: What is the "glaciological method"?

We will extend the text as follows: The seasonal surface mass balance of Triangular Glacier was estimated using a glaciological method, spatial interpolation of melt and accumulation measured at ablation stakes, once a year for the years 2014/15–2019/20.

Lines 120-21: Why was surface mass balance only estimated for 2014-2020? These study periods don't match up across the entire study and needs to be addressed.

The mass balance measurements were only carried out in 2014-2020. In combination with the volume measurements in 2006 and 2015, we were able to simulate the mass balance for the whole study period (L121-124). Please see also our answer to your comment about Lines 304 – 305.

Lines 125- 127: The section on snow depth needs more detail -- was snow depth only measured at one location? How was it extrapolated across the catchment? Is the snow on the glacier the only thing assumed to contribute to runoff? What about snow in the rest of the catchment? Was SWE/snow density measured as well?

Snow depth was only measured at one site. This data was not used to calibrate the model. It was only used as an additional check that the model simulated a realistic snow cover. No SWE measurements are available from this area.

We will modify this section as follows: "Snow depth was measured using a sonic distance sensor (Judd Communication, USA) fixed at the automatic weather station located in the central part of Triangular Glacier during the period 6 February 2017 to 23 January 2020. The zero point level was chosen arbitrarily (Engel et al., 2023). The snow depth was only used to calculate the estimated SWE, which was calculated as the observed snow depth multiplied by constant density 389 kg·m-3 which was the mean measured density in summer season 2022. The estimated SWE time series was only used for model evaluation".

Additionally, we will replace "snow density" with "estimated SWE" throughout the manuscript.

Lines 132: model routines need to be defined. Were all of these routines used? Or was one chosen from these 5?

All of these routines were used since only together they can calculate the water propagation through the entire system and simulate the runoff in the outlet. As we noted in the response to one of your comments above, we realized that the model description, despite referenced literature, is not fully clear and some important details are missing. We have decided to extend the methods section substantially to better explain how the model deals with input data and how the individual water balance components are calculated.

Lines 146: Why was this section not considered in the model?

Unfortunately, no measurements of the mass balance from this area are available. However, the small part of the ice cap is located in the highest part of the catchment. Therefore, we can assume that this part does not contribute with glacier-melt water to runoff and thus can be technically considered as "glacier-free" in the model (which contributes only by occasional snowmelt to the total runoff as explained on L328-331 in the original manuscript). This our assumption was later supported by model results where no glacier melt was simulated above approximately 350 m a.s.l. We will consider reformulation of the existing text both in methods and discussion for clarity.

Lines 155-165: The runoff is reported in mm, but I assume it was measured in some rate (i.e. l/s). How was this rate converted to mm and why? Was it summed up over the daily, monthly, yearly period?

Yes, the streamflow (in $m^3\ s^{-1}$) was calculated from direct water level measurements using a rating curve. However, it is a common (a good) praxis in hydrology and hydrological modelling to report streamflow as runoff depth (the term "runoff" is frequently used), which is a water volume per unit area (usually catchment) for a defined period (daily, monthly, annual) or a long-term mean (therefore the units are mm). Among others, this approach enables the direct comparison of all measured and simulated water balance components (precipitation, PET, snow, glacier, soil and groundwater storage) available in the same units.

We will go throughout the whole text to ensure that all values and their sums (daily to annual or a mean) are always clearly reported and named by correct terminology.

1. RESULTS:

Lines 168 -171: These values could be put into a table to show the overall model fit.

As stated in the methods, we performed 100 calibration runs to address equifinality and increase the model robustness. The range of the objective function values is shown in Fig. A1. We will consider of creating a simple table showing the median values which are now presented only in the main text. Furthermore, we will fix the typo (S1 to A1) and move the reference to the second sentence of this paragraph where it makes more sense.

Lines 176-177: Glacier mass balance simulation and observed should be included in Figure 2.

When writing manuscript, we decided to move some of the plots showing the model performance in the appendix. This also covers the comparison of simulated and observed mass balance which we placed as Fig. A2 in the original manuscript. However, we will consider putting all these plots directly into the results section.

Lines 179-180: There was never a mention of SWE or snow density measurements in the methods. How was simulated SWE compared to snow depth? How did the model estimate SWE and snow depth? This needs a lot more detail in the methods and results.

We refer to our response to your comment on Lines 125- 127 above. Additionally, we will add the following text to the respective section: "During validation, mean simulated SWE in the elevation zone, in which the weather station is located was compared to estimated SWE."

Figure 3: The cumulative line in (a) should be a color that is easier to see (or does this line really need to be there?). The axes also need more values, especially at 0 in (a) to see mass loss or gain.

*We agree, we will modify the image to make it easier to read.*

Lines 197: the glacier did not melt at all? Might be more appropriate to say "did not lose mass" as there was likely still some melt generated on the glaciers even if they did not lose mass.

*We will rephrase the sentence as you suggested.*

Lines 198: Are these simulated SWE values? SWE measurements were never addressed in the methods section so I'm unsure if these values are from the model or measurements. If they are from the model, this needs to be added to the methods section of how the model calculates and estimates SWE.

*These are simulated SWE values. We will add (to the method section) that measured SWE was only used for model evaluation. Please also see our previous response to your related comments above.*

Lines 202: What does it mean "these months accounted for 32% and 25% of the total SWE loss"? This needs to be explained more and likely shown in a figure.

*We agree that this specific result is not sufficiently described and might be confusing. Instead of this, we will consider adding a new figure or table showing the fraction of the mean monthly snowmelt on the mean annual snowmelt. Please see also a simple plot showing the values (we will improve the visual look and clarity of the plot if we decide to include it in the revised version).*

[Figure]

*Fig. Mean monthly snowmelt fractions to the mean annual snowmelt*

Lines 204: The simulated mean runoff was 415 mm? Should this be 415 mm per year?

*As stated in the sentence, we refer to "mean annual runoff".*

Lines 204-205: What were the instances when there was runoff outside of the main summer season? When did this winter runoff occur? This is brought up in the abstract and conclusion and needs much more attention in the results and discussion.

We agree that this particular result is not well addressed in the discussion section. We will add the following text to the discussion to clarify this issue: "According to Kavan et al., (2023) seven studies have been devoted to measuring runoff in the Antarctic Peninsula area between 1991 and 2020(Inbar, 1995; Kavan et al., 2017; Sziło and Bialik, 2017; Falk et al., 2018; Stott and Convey, 2021; Kavan, 2021; Kavan et al., 2023) For most of the studies, the duration of the measurements did not exceed two months. Only one study (Stott and Convey, 2021) measured for 78 days. The measurements were mostly made during January and February, sometimes also in March. Only one study (Stott and Convey, 2021) covered most of December. However, no study took measurements during September, October, November or May. Therefore, our results, which include 11 years of simulated daily runoff, provide valuable information, although they are subject to the uncertainties of modelling based on small amounts of data."

Lines 205: How was evapotranspiration estimated? This was never mentioned before in the methods.

The method of calculation of daily potential evapotranspiration is mentioned on line 104. The actual evapotranspiration was calculated by the model as a function of simulated soil moisture and air temperature. We will provide a better description of the model routine along with other model routine descriptions as mentioned in our responses above.

Lines 206: It was never stated how much of the precipitation was estimated to come from snow vs. rain, or how the model partitions this. So how was the different contributions estimated between rain and snow in the model?

The HBV model is partitioning snowfall and rainfall using a concept of a single threshold temperature ($T_T$). The $T_T$ is one of the model parameters calibrated by the model. Since the catchment is divided into elevation zones, the precipitation phase is then calculated separately for the specific elevation zone using a calibrated temperature lapse rate. Further, the model keeps track of whether the water comes from rain or snowfall (or glacier melt) throughout the other model components until leaving the catchment, assuming the complete mixing of water in individual components (so-called effect tracking; Weiler et al., 2018).

Similar to your comments and our responses above, we will include better explanation of precipitation partitioning along with more detailed description of model structure and parameters.

Figure 4: The axes should be colored for each of the values they are representing, it is confusing as is. Also, these should be split into two panels so that the various y axes don't overlap. Precip and temp should be panel A with only one y-axis on each side. Panel B below should be Q and cumulative Q.

We will modify the image to make it easier to read.

Figure 5: can these bars be stacked to add up to total Q instead of being side by side? That will make it easier to see how the allocation of various Q contributions changes.

Thank you for the suggestion. We tried to redesign the figure following the reviewer's suggestion, but we believe that the original form is clearer and allows easier comparison between runoff components between months. Nevertheless, we will try to find other solutions to increase readability and select one we will consider as best.

Figure 6: Similar to Fig. 5 these figures are confusing and have too many axis values. It would be beneficial to split into panels so that the axes labels do not overlap.

We will modify the image to make it easier to read.

Line 247: is this referring to snow cover on the glacier that melted out earlier?

Yes, we will rephrase the sentence for clarity.

Line 276: I am confused how Qglacier can be the only one significantly correlated with Q total, and how it can have the dominant control over total runoff, but Qsnow dominates the proportion of total runoff (over 75%)? This needs more in depth analysis and thought into what this means in both the results and discussion section.

Of the three runoff components, annual $Q_{glacier}$ has the highest inter-annual variability (see the table below). This is because it is more dependent on annual temperature than $Q_{snow}$ and $Q_{rain}$. For snow, as an example, it means that there is a similarly equal amount of snow in individual years, which results in low inter-annual variability of snow runoff contribution even though it is a dominant component of the total runoff. Therefore, the inter-annual variations of the total runoff are dominantly caused by inter-annual variations in glacier-melt runoff rather than snowmelt runoff.

We agree that the text in the original manuscript is not fully clear in this respect, therefore we will reformulate it.

*Table: Coefficient of variation for the individual runoff components*

| Annual Q | Annual $Q_{rain}$ | Annual $Q_{snow}$ | Annual $Q_{glacier}$ |
|---|---|---|---|
| 0.15 | 0.53 | 0.16 | 0.79 |

Figure 9: It is hard to see the solid vs dotted circles, may want to make the outlines bigger or different colors.

Thank you for the suggestion, we will modify the image to make it easier to read.

1. DISCUSSION:

Lines 304 – 305, should address **the discrepancies in study periods for all of the data and how this may affect the interpretation of results**.

Thank you for the comment. Besides better clarification of the input data and time periods in the methods section (as mentioned above), we will add the following text to the discussion section:

"It must be noted that the model calibration and simulation were carried out on different data sets, which, except for air temperature, precipitation and PET, did not cover the whole study period. The streamflow measurements were only taken during one summer season (8 February–15 March 2018) and may not fully represent the variability of streamflow during the whole study period. The mass balance measurements were carried out in 2014-2020, which in combination with the glacier volume measurements in 2015/16 gave us sufficient information for the model calibration period. However, the volume of Triangular Glacier at the start of the simulation in 2009 was estimated from a digital elevation model of the ice surface of Triangular Glacier in 2006, assuming a constant change in glacier extent of -0.3% per year as stated in (Engel et al., 2023) as the mean annual retreat rate between 2006 and 2014. For this reason, the uncertainty of the initial state of the glacier was introduced into the simulation."

Lines 316: Where did these percip measurements come from? From the methods, I thought that there were no precip observations so this needs to be addressed earlier. Are there measurements of both rain and snow?

We agree that the respective section might be confusing. Therefore, we will add the following text to the method section: "Short-term precipitation measurements were taken in January and February 2022 at the Johann Gregor Mendel Station using a Thies laser precipitation monitor (disdrometer) and a manual rain gauge.  These data were used to validate the performance of the Weather Research and Forecasting (WRF) model (Matějka et al., 2022)."

Lines 326: So SWE was measured? Needs to be brought up in methods.

Please see our answers to your comments on Lines 125-127 and Lines 179-180.

Lines 331: I thought this part was not considered in the model at all, but now it is saying that snow from this area was considered? Why is this? This needs to be explained more.

We will rephrase the text in methods so it is clear that the highest part of the catchment is considered as glacier-free area in the model. Please, see also our response related to your comment on Line 146.

Lines 333 – 334: The statement "the glacier is covered with snow most of the year preventing any potential glacier melt" needs to be backed up. How does snow prevent glacier melt? Is there data to show that the glacier is actually covered with new snow most of the year? If there is no data or references to back this up than this statement needs to be removed.

Thank you for the comment. After reviewing our text we decided to remove the respective sentence since it is not necessary.

Lines 349 – 350: What are the different definitions of the start year?  Why was June 1 chosen for this study?

Thank you for the comment. After reviewing our text we decided to remove the respective sentence since it is not necessary. Additionally, we will reformulate the whole section for clarity.

Lines 351-355: Why is this statement worth mentioning? If the authors feel that this is important to mention then this needs to be explained and elaborated on further here.

We agree that this sentence is not necessary. We will remove it.

Lines 364-365: It is also important to note that the flow season for streams in the McMurdo Dry Valleys is much shorter than on the Peninsula, and most of the streamflow comes from glacial melt.

Thank you for the suggestion, we will reformulate the text as follows: "The reason for this finding may be that the total precipitation amount is lower compared to our study area, resulting in lower snow accumulation on glaciers and thus its earlier melt-out, causing the earlier initiation of glacier melt, which is the primary source of runoff. It is important to note that the flow season in the McMurdo Dry Valleys is shorter than that observed in the study area, with an average length of 70 days (Gooseff and Lyons, 2007)."

Lines 370: AP?

We will change it to "Antarctic Peninsula"

Lines 371 -372: The statement "this in turn may cause water shortages and affect local freshwater ecosystems" can be explained much more. How does water availability in these areas affect the ecosystems? Why is it important to study the runoff regimes of these areas for the environment and ecosystems? This needs references and elaboration to back it up.

Thank you for the comment. We will consider reformulating the whole section in lines 371-376 and include more references to make it more informative.

Lines 373-374: How does rising temperatures influence suspended sediment transport? And what does this mean for this study? This section of the discussion needs to be expanded on quite a bit for the broader environmental implications.

Thank you for the comment. We will consider reformulating the whole section in lines 371-376 and include more references to make it more informative.

Lines 376: Again, needs more expansion on the idea that higher amounts of rainfall can affect melting of snow and glaciers. How? And why is this important? Need references to back this idea up.

Thank you for the comment. We will consider reformulating the whole section in lines 371-376 and include more references to make it more informative.

CONCLUSIONS:

Line 391 – 392: the statement: "Together with mean annual evapotranspiration (7 mm) and mean annual precipitation (369 mm)." is not a complete sentence, maybe combine it with the previous sentence.

Thank you for pointing out that. We will correct the sentence.

Lines 396: Again, what does "strong glacier and snow melt" mean?

We agree that the word "strong" is too vague and redundant in the sentence. We will remove it from the text.

Lines 398-399: as stated before, I am confused how Qglacier can be the only control on total annual Q, but Qsnow is the main contributor to Qtotal. This needs to be clarified in the discussion and if stated in the conclusion then it needs to be explained more.

Please see our response to your comment on Line 276. We will reformulate the sentence for clarity.

Lines 404-405: The presence of runoff-generating events outside of the summer season was not addressed in detail in the discussion section, and if it is going to be included in the conclusion and abstract it needs much more attention and analysis outside of the brief mention in results.

Please see our response to your comment on Lines 204-205. We will also consider reformulation here for clarity.

**References**

Carrasco, J. F. and Cordero, R.: Analyzing Precipitation Changes in the Northern Tip of the Antarctic Peninsula during the, 2020.

Weiler, M., Seibert, J., and Stahl, K.: Magic components—why quantifying rain, snowmelt, and icemelt in river discharge is not easy, Hydrol. Process., 32, 160–166, https://doi.org/10.1002/hyp.11361, 2018.

---

## Author Comment (AC2)

Thank you for the review of our manuscript. We appreciate your constructive comments and suggestions. Please find our point-by-point response below (in blue).

The manuscript is well written and good to publish in the journal. My main concern is about the degree-day factors used in the study. Therefore, my specific comments are as follows:

Line 132-133: It is good that the degree-day-based snow melt and glacier melt modules were used in the HBV model. It is necessary to mention whether the degree-day factors are model-calibrated or assigned values derived from the field measurement in the past. Since the degree-day factors play a significant role in the ablation estimation, it should be mentioned degree-day factors in this study. Moreover, sometimes model-generated degree-day factors may be unrealistic. Therefore, I suggest mentioning degree-day factors used in this study.

Based on this comment and also the one from the second reviewer we realized that the method section related to the model structure and parameters needs to be extended since a lot of information is missing there (despite references to other literature). This also applies to a better explanation of the snow routine of the model which is based on the degree-day method.

The degree-day factors both for snow and glaciers are one of the model parameters and thus they were calibrated. Before the calibration, the upper and lower limits have been applied to ensure that parameters are physically relevant. In the case of our study, the lower limit was set to 2 mm $°C^{-1}$ $d^{-1}$ and the upper limit to 8 mm $°C^{-1}$ $d^{-1}$. However, after leaving the same range for non-glaciated and glaciated parts of the catchment, the model simulated high snow accumulations at the highest elevation zones of the catchment which did not completely melt in the season and thus created an unrealistic increase in snow storage over the study period (in hydrological modelling literature often referred to as "snow towers"). Therefore, we needed to do a fine-tuning of the degree-day factors separately for non-glaciated (median value resulting from 100 calibration runs was 6.01 mm $°C^{-1}$ $d^{-1}$) and glaciated parts of the catchment (median value equal to 2.23 mm $°C^{-1}$ $d^{-1}$) to achieve realistic snow storage simulations.

Additionally, we provided a simple evaluation of the simulated snow water equivalent and snow depth (measured automatically during a single summer season) as described in L179-182 and Appendix A2 of the original manuscript. Despite the lack of data, the modelled SWE values correlated well with measured ones.

We will reformulate the methods section on this topic reflecting the above explanation and include the calibrated degree-day factor values.

Line 347: It should be …. was also slightly negative in 2014/15, ……………..

Thank you, we will correct the sentence.

Line 435: Missing the reference of Seibert and Vis (2012).

We are not sure whether we interpret this comment correctly since the referenced line (L435) represents the beginning of the references section. The mentioned reference is included in the reference list (L607).

---

## Author Response (AR1)

**Review 1**

Thank you for the review of our manuscript. We appreciate your constructive comments and suggestions. Please find our point-by-point response below (in blue). The line numbering in our response refers to the revised manuscript.
In general, we have identified the following main issues, which we have addressed below:

1) We found the method section related to a description of the HBV model (structure, routines, parameters) too brief and we have described it in more detail.
2) We have been more explicit about the data used for model calibration and simulation. In particular, we have clarified how the discharge, glacier mass balance and snow data have been used to model calibration and evaluation
3) We have better explained how the runoff components were calculated in the model and how to interpret the results related to the inter-annual variability of the components and explain the contradiction between the snowmelt which contributes most to the total runoff and glacier-melt which controls the inter-annual variability.
4) More emphasis has been placed on a literature review and discussion related to understanding the runoff process in polar areas and its impact on terrestrial ecosystems.

Overall, this manuscript is within the scope of the journal and adds some significant contributions to our understanding of the water budget and runoff processes on the Antarctic Peninsula. As I understand it, this paper is about quantifying the various contributions from snow, glacial melt, and rain to streamflow on the Antarctic Peninsula, and the relationships of total runoff and runoff contributions to climate variations. However, the title only mentions snow and glacial contributions but the paper also addresses rain contributions as well. There is also some confusion with the various study periods for the different model components as well as the field validations and measurements. The methods section needs more detail on the model: how the model uses the input data and how it estimates many of the simulated values, and also the field validation methods and what exactly was measured and how it was used. The discussion section also needs to be expanded in order to elaborate on the importance of the results and the implications for the ecosystems in this region and the effects of climate on runoff processes.

**Specific comments**

The title only mentions snow and glacial contributions but the paper also addresses rain contributions as well.

When formulating the research questions, we were interested mostly in snow and glacier contributions since the rainfall is rather minor. However, we agree that the original title might be confusing, so we have decided to change the title accordingly to "The role of snowmelt, glacier melt and rainfall in streamflow dynamics on James Ross Island, Antarctic Peninsula"

1. ABSTRACT AND INTRODUCTION:

Lines 13-14: It would be helpful to actually state what the total study period is (i.e. June 2010 – May 2021).

We agree that this change might improve readability. We reformulated the respective text to "We used the hydrological model HBV to simulate the runoff process from June 2010 to May 2021 at a daily resolution." on lines 14-15.

Lines 17: What does "strong glacier and snow melt" mean?

We found the word "strong" to be too vague and redundant in the sentence. We removed it from the text on line 18.

Lines 25: What are "high air temperatures in recent years"? Can this be quantified above average or above previous maximum temperatures?

We agree that the text is too vague. We rephrased it to "Although the warming of this region was interrupted in the early part of the 21st century (Turner et al., 2016; Oliva et al., 2017), the cooling period ended in the mid-2010s and (Carrasco et al., 2021) since then the region has experienced several warm events with record high temperatures (González-Herrero et al., 2022)." (lines 26-28).

Line 40: It may be helpful to also note that all of the precipitation in the dry valleys is in the form of snow (whereas the peninsula also receives some rain, analyzed in this paper).

We added the note to the text on lines 42-44: "In addition, this region receives a certain amount of precipitation in the form of rain, unlike the Dry Valleys where only snowfall was observed (Fountain et al., 2010).".

Line 55 and Line 62: It would be helpful to include some description of the terrestrial and marine environments in these regions in relation to streamflow and how they are affected by these runoff processes. What is the importance of understanding runoff processes for these ecosystems and environments?

The description of the impact of changes in terrestrial ecosystems on streamflow is stated in L30-35. In the revised version, we additionally rephrased the text along lines 57-59 to be more specific.

Lines 66: Again the study period of 2010/2011 – 2020/2021 is confusing. It may be helpful to state that streamflow only occurs during XX months, so the study covers the time period of June 2010 – May 2021 or something along those lines…

We agree and we reformulated the text on line 69.

1. METHODS:

Lines 86-87: How much of the precipitation is estimated to come from rainfall and from snow each year? Has this changed over time?

As previously stated on L106, numerical models represent the only available source of information about precipitation from a long-term perspective in the study area. However, data from three stations located west of the Antarctic Peninsula Carrasco and Cordero, (2020) indicate an increase in precipitation from 1970 to the early 1990s, followed by a decrease from 1991 to 1999. Additionally, an increase in snowfall and a decrease in rainfall were

observed from the mid-1990s to the mid-2010s, resulting from the cooling period. We added this information to the text on lines 90-93.

Lines 70 and on… Some further discussion of the streams where discharge is being measured would be helpful. How much does the discharge vary from year to year? What is the average flow season? Is runoff only estimated at one location/stream?

Indeed, in such remote areas, the water level is not possible to measure continuously over longer periods. Further uncertainties arise from discharge calculation using rating curves which need to be created separately for each year due to the riverbed changes. Usually, the discharge data is only available from a few locations around the polar station and only during its operation (usually one to two months each year). For our study area of the Triangular catchment, the only available measurements cover the time period from February to March 2018 which was used as one of the variables for the HBV model calibration (see the respective lines 177-178). Therefore, neither the inter-annual variability nor the length of the discharge season are known. However, based on your comment, we decided to extend the discussion on the above topic on lines 347-355.

Line 99: Why was 1 June estimated as the start of the water year?

After a careful literature search, we are not aware of any systematically used definition of the hydrological year in this area. Therefore, we determined the beginning of the hydrological year based on our data. We have chosen June as the beginning because (based on our dataset) from that month onwards the number of days with precipitation and air temperature above 0 °C is negligible.

Line 99: Were air temp and total precip the only inputs for the model? How does it use those to estimate runoff and glacier mass balance?

The model inputs are formed by air temperature (T), precipitation (P) and potential evapotranspiration (PET; calculated using air temperature). Available data of glacier mass balance and runoff was used for model calibration. The whole procedure is described in section 2.3. However, we realized that the model description was not fully clear in the original manuscript and some important details were missing (e.g. the better description of snowmelt and glacier routines), although references to detailed model descriptions were provided in the original text. We significantly extended the methods section to better explain how the model deals with input data and how the individual water balance components are calculated. Therefore, we modified the whole section 2.3.

Lines 106: Was there any field validation for the simulated precipitation?

Yes, there was a field validation of the simulated precipitation. We added the lines 111-113 and 121-122.

Lines 116: Why was the runoff only calculated from Feb – Mar 2018? Was there no other data outside this period? How did this period relate to the rest of the study period?

Please also see our response to one of your similar comments above. There was no other measured discharge data in the study catchment outside this period. We are aware that this period represents only a small part of the entire study period and this was one of the reasons

why we used glacier mass balance measurements for model calibration. As noted above we extended the discussion section on lines 347-355 related to the topic with a new paragraph (which also relates to your comment on lines 304-305 of the original manuscript).

Lines 120: What is the "glaciological method"?

We believe this is a well-established method and it is beyond the scope of this paper to describe it. We have therefore included a reference to a paper describing this method (line 127).

Lines 120-21: Why was surface mass balance only estimated for 2014-2020? These study periods don't match up across the entire study and needs to be addressed.

The mass balance measurements were only carried out in 2014-2020. In combination with the volume measurements in 2006 and 2015, we were able to simulate the mass balance for the whole study period (L127-132). Please see also our answer to your comment about Lines 304 – 305.

Lines 125- 127: The section on snow depth needs more detail -- was snow depth only measured at one location? How was it extrapolated across the catchment? Is the snow on the glacier the only thing assumed to contribute to runoff? What about snow in the rest of the catchment? Was SWE/snow density measured as well?

Snow depth was only measured at one site. This data was not used to calibrate the model. It was only used as an additional check that the model simulated a realistic snow cover. No SWE measurements are available from this area. We modified the lines 133-137 accordingly. Additionally, we replaced "snow density" with "estimated SWE" throughout the manuscript.

Lines 132: model routines need to be defined. Were all of these routines used? Or was one chosen from these 5?

All of these routines were used since only together they can calculate the water propagation through the entire system and simulate the runoff in the outlet. As we noted in the response to one of your comments above, we realized that the model description, despite the referenced literature, was not fully clear in the original manuscript and some important details were missing. We have decided to extend the methods section 2.3 significantly to better explain how the model deals with input data and how the individual water balance components are calculated.

Lines 146: Why was this section not considered in the model?

Unfortunately, no measurements of the mass balance from this area are available. However, the small part of the ice cap is located in the highest part of the catchment. Therefore, we can assume that this part does not contribute with glacier-melt water to runoff and thus can be technically considered as "glacier-free" in the model (which contributes only by occasional snowmelt to the total runoff as explained on lines 385-390). This assumption was later supported by model results where no glacier melt was simulated above approximately 350 m a.s.l. Besides reformulating the discussion section in this respect, we also better specified it in the methods section on lines 166-167.

Lines 155-165: The runoff is reported in mm, but I assume it was measured in some rate (i.e. l/s). How was this rate converted to mm and why? Was it summed up over the daily, monthly, yearly period?

Yes, the streamflow (in $m^3\ s^{-1}$) was calculated from direct water level measurements using a rating curve. However, it is a common (and good) praxis in hydrology and hydrological modelling to report streamflow as runoff depth (the term "runoff" is frequently used), which is a water volume per unit area (usually catchment) for a defined period (daily, monthly, annual) or a long-term mean (therefore the units are mm per defined period). Among others, this approach enables the direct comparison of all measured and simulated water balance components (precipitation, PET, snow, glacier, soil and groundwater storage) available in the same units. We went throughout the whole text to ensure that all values and their sums (daily to annual or a mean) are always clearly reported and named by correct terminology.

1. RESULTS:

Lines 168 -171: These values could be put into a table to show the overall model fit.

As stated in the methods, we performed 100 calibration runs to address equifinality and increase the model robustness. The range of the objective function values was shown in the supplement as Fig. A1 in the original manuscript. However, we decided to show the figures in the main text as Figure 2 and thus bring them closer to readers to avoid jumping between the supplement and the main text. Therefore, we think the additional table showing the same is not needed.

Lines 176-177: Glacier mass balance simulation and observed should be included in Figure 2.

Similar to our response above, when writing the manuscript, we decided to move some of the plots showing the model performance to the supplement. However, we newly have merged Fig. A2 and Fig. 3, so the comparison of simulated and observed values is now in the results section.

Lines 179-180: There was never a mention of SWE or snow density measurements in the methods. How was simulated SWE compared to snow depth? How did the model estimate SWE and snow depth? This needs a lot more detail in the methods and results.

We refer to our response to your comment on Lines 125- 127 above. Additionally, we added the following text on lines 181-182: "During validation, the mean simulated SWE in the elevation zone, in which the weather station is located was compared to estimated SWE."

Figure 3: The cumulative line in (a) should be a color that is easier to see (or does this line really need to be there?). The axes also need more values, especially at 0 in (a) to see mass loss or gain.

We agree, we modified the image to make it easier to read.

Lines 197: the glacier did not melt at all? Might be more appropriate to say "did not lose mass" as there was likely still some melt generated on the glaciers even if they did not lose mass.

We rephrased the sentence as suggested (line 232).

Lines 198: Are these simulated SWE values? SWE measurements were never addressed in the methods section so I'm unsure if these values are from the model or measurements. If they are from the model, this needs to be added to the methods section of how the model calculates and estimates SWE.

These are simulated SWE values. We modified the method section on lines 136-137 to clearly state that measured SWE was only used for model evaluation. Please also see our previous response to your related comments above.

Lines 202: What does it mean "these months accounted for 32% and 25% of the total SWE loss"? This needs to be explained more and likely shown in a figure.

We agree that this specific result was not sufficiently described and might be confusing. We have modified the lines L236-237 and added table 2.

Lines 204: The simulated mean runoff was 415 mm? Should this be 415 mm per year?

As stated in the sentence, we refer to "mean annual runoff".

Lines 204-205: What were the instances when there was runoff outside of the main summer season? When did this winter runoff occur? This is brought up in the abstract and conclusion and needs much more attention in the results and discussion.

We agree that this particular result is not well addressed in the discussion section. We have added the following text to the discussion (lines 405-412) to clarify this issue: "According to Kavan et al. (2023) seven studies have been devoted to measuring runoff in the Antarctic Peninsula area between 1991 and 2020 (Inbar, 1995; Kavan et al., 2017; Sziło and Bialik, 2017; Falk et al., 2018; Stott and Convey, 2021; Kavan, 2021; Kavan et al., 2023). For most of the studies, the duration of the measurements did not exceed two months. Only one study (Stott and Convey, 2021) measured for 78 days. The measurements were mostly made during January and February, sometimes also in March. Only one study (Stott and Convey, 2021) covered most of December. However, no study took measurements during September, October, November or May. Therefore, our results, which include 11 years of simulated daily runoff, provide valuable information, although they are subject to the uncertainties of modelling based on limited amounts of data."

Lines 205: How was evapotranspiration estimated? This was never mentioned before in the methods.

The method of calculation of daily potential evapotranspiration is mentioned on lines 143-145. The actual evapotranspiration was calculated by the model as a function of simulated soil moisture and air temperature. We have specified it better on line 149.

Lines 206: It was never stated how much of the precipitation was estimated to come from snow vs. rain, or how the model partitions this. So how was the different contributions estimated between rain and snow in the model?

The HBV model is partitioning snowfall and rainfall using the concept of a single threshold temperature ($T_T$). The $T_T$ is one of the model parameters calibrated by the model. Since the catchment is divided into elevation zones, the precipitation phase is then calculated separately for the specific elevation zone using a calibrated temperature lapse rate. Further, the model keeps track of whether the water comes from rain or snowfall (or glacier melt) throughout the other model components until leaving the catchment, assuming the complete mixing of water in individual components (so-called effect tracking; Weiler et al., 2018). Similar to your comments and our responses above, we included a better explanation of precipitation partitioning along with a more detailed description of the model structure and parameters in section 2.3, mainly on lines 153-173.

Figure 4: The axes should be colored for each of the values they are representing, it is confusing as is. Also, these should be split into two panels so that the various y axes don't overlap. Precip and temp should be panel A with only one y-axis on each side. Panel B below should be Q and cumulative Q.

We have modified the image to make it easier to read.

Figure 5: can these bars be stacked to add up to total Q instead of being side by side? That will make it easier to see how the allocation of various Q contributions changes.

Thank you for the suggestion. We tried to redesign the figure following the reviewer's suggestion, but after careful consideration, we came back to the original version which we believe is clearer and allows easier comparison between runoff components in individual months.

Figure 6: Similar to Fig. 5 these figures are confusing and have too many axis values. It would be beneficial to split into panels so that the axes labels do not overlap.

We have modified the image to make it easier to read.

Line 247: is this referring to snow cover on the glacier that melted out earlier?

It refers to the mean snow cover in the catchment including the snow cover on the glacier.

Line 276: I am confused how Qglacier can be the only one significantly correlated with Q total, and how it can have the dominant control over total runoff, but Qsnow dominates the proportion of total runoff (over 75%)? This needs more in depth analysis and thought into what this means in both the results and discussion section.

Of the three runoff components, annual $Q_{glacier}$ has the highest inter-annual variability (see newly added Table 4). This is because it is more dependent on annual temperature than $Q_{snow}$ and $Q_{rain}$. For snow, as an example, it means that there is a similarly equal amount of snow in individual years, which results in low inter-annual variability of snow runoff contribution even though it is a dominant component of the total runoff. Therefore, the inter-annual variations of the total runoff are dominantly caused by inter-annual variations in glacier-melt runoff rather than snowmelt runoff.

We agree that the text in the original manuscript was not fully clear in this respect, therefore we have reformulated it on lines 319-321 and added new Table 4.

Figure 9: It is hard to see the solid vs dotted circles, may want to make the outlines bigger or different colors.

Thank you for the suggestion, we modified the image to make it easier to read.

1. DISCUSSION:

Lines 304 – 305, should address the discrepancies in study periods for all of the data and how this may affect the interpretation of results.

Thank you for the comment. Besides better clarification of the input data and time periods in the methods section (as mentioned above), we have added lines 347-355.

Lines 316: Where did these percip measurements come from? From the methods, I thought that there were no precip observations so this needs to be addressed earlier. Are there measurements of both rain and snow?

We agree that the respective section might be confusing. Therefore, we added the following text to the method section on lines 111-113: "Short-term precipitation measurements were taken in January and February 2022 at the Johann Gregor Mendel Station using a Thies laser precipitation monitor (disdrometer) and a manual rain gauge."

Lines 326: So SWE was measured? Needs to be brought up in methods.

Please see our answers to your comments on Lines 125-127 and Lines 179-180.

Lines 331: I thought this part was not considered in the model at all, but now it is saying that snow from this area was considered? Why is this? This needs to be explained more.

We rephrased the text in methods (lines 165-167) so it is clear that the highest part of the catchment is considered as glacier-free area in the model. Please, see also our response related to your comment on Line 146.

Lines 333 – 334: The statement "the glacier is covered with snow most of the year preventing any potential glacier melt" needs to be backed up. How does snow prevent glacier melt? Is there data to show that the glacier is actually covered with new snow most of the year? If there is no data or references to back this up than this statement needs to be removed.

Thank you for the comment. After reviewing our text we have decided to remove the respective sentence since it is not necessary.

Lines 349 – 350: What are the different definitions of the start year?  Why was June 1 chosen for this study?

Thank you for the comment. After reviewing our text we decided to remove the respective sentence since it is not necessary.

Lines 351-355: Why is this statement worth mentioning? If the authors feel that this is important to mention then this needs to be explained and elaborated on further here.

We agree that this sentence is not necessary. We removed it.

Lines 364-365: It is also important to note that the flow season for streams in the McMurdo Dry Valleys is much shorter than on the Peninsula, and most of the streamflow comes from glacial melt.

Thank you for the suggestion, we reformulated the text (lines 423-425).

Lines 370: AP?

We changed it to "Antarctic Peninsula".

Lines 371 -372: The statement "this in turn may cause water shortages and affect local freshwater ecosystems" can be explained much more. How does water availability in these areas affect the ecosystems? Why is it important to study the runoff regimes of these areas for the environment and ecosystems? This needs references and elaboration to back it up.

Thank you for the comment. We reformulated the whole section in lines 434-444.

Lines 373-374: How does rising temperatures influence suspended sediment transport? And what does this mean for this study? This section of the discussion needs to be expanded on quite a bit for the broader environmental implications.

Thank you for the comment. We reformulated the whole section in lines 434-444.

Lines 376: Again, needs more expansion on the idea that higher amounts of rainfall can affect melting of snow and glaciers. How? And why is this important? Need references to back this idea up.

Thank you for the comment. We reformulated the whole section in lines 434-444.

CONCLUSIONS:

Line 391 – 392:  the statement: "Together with mean annual evapotranspiration (7 mm) and mean annual precipitation (369 mm)." is not a complete sentence, maybe combine it with the previous sentence.

Thank you for pointing out that. We corrected the sentence on Line 461.

Lines 396: Again, what does "strong glacier and snow melt" mean?

We agree that the word "strong" is too vague and redundant in the sentence. We have removed it from the text (line 466).

Lines 398-399: as stated before, I am confused how Qglacier can be the only control on total annual Q, but Qsnow is the main contributor to Qtotal. This needs to be clarified in the discussion and if stated in the conclusion then it needs to be explained more.

Please see our response to your comment on Line 276. We reformulated the sentence for clarity (lines 468-469).

Lines 404-405: The presence of runoff-generating events outside of the summer season was not addressed in detail in the discussion section, and if it is going to be included in the conclusion and abstract it needs much more attention and analysis outside of the brief mention in results.

Please see our response to your comment on Lines 204-205. We have reformulated the sentence on line 475.

**References**

Carrasco, J. F. and Cordero, R.: Analyzing Precipitation Changes in the Northern Tip of the Antarctic Peninsula during the, 2020.

Weiler, M., Seibert, J., and Stahl, K.: Magic components—why quantifying rain, snowmelt, and icemelt in river discharge is not easy, Hydrol. Process., 32, 160–166, https://doi.org/10.1002/hyp.11361, 2018.

**Review 2**

Thank you for the review of our manuscript. We appreciate your constructive comments and suggestions. Please find our point-by-point response below (in blue). The line numbering in our response refers to the revised manuscript.

The manuscript is well written and good to publish in the journal. My main concern is about the degree-day factors used in the study. Therefore, my specific comments are as follows:

Line 132-133: It is good that the degree-day-based snow melt and glacier melt modules were used in the HBV model. It is necessary to mention whether the degree-day factors are model-calibrated or assigned values derived from the field measurement in the past. Since the degree-day factors play a significant role in the ablation estimation, it should be mentioned degree-day factors in this study. Moreover, sometimes model-generated degree-day factors may be unrealistic. Therefore, I suggest mentioning degree-day factors used in this study.

Based on this comment and also several others from the second reviewer we realized that the method section related to the model structure and parameters needs to be extended since a lot of information was missing in the original manuscript (despite references to other literature). This also applies to a better explanation of the snow routine of the model which is based on the degree-day method.

The degree-day factors both for snow and glaciers are one of the model parameters and thus they were calibrated. Before the calibration, the upper and lower limits have been applied to ensure that parameters are physically relevant. In the case of our study, the lower limit was set to 2 mm °C$^{-1}$ d$^{-1}$ and the upper limit to 8 mm °C$^{-1}$ d$^{-1}$. However, after leaving the same range for non-glaciated and glaciated parts of the catchment, the model simulated high snow accumulations in the highest elevation zones of the catchment which did not completely melt in the season and thus created an unrealistic increase in snow storage over the study period (in hydrological modelling literature often referred to as "snow towers"). Therefore, we needed to do a fine-tuning of the degree-day factors separately for non-glaciated (median value resulting

from 100 calibration runs was 6.01 mm $°C^{-1}$ $d^{-1}$) and glaciated parts of the catchment (median value equal to 2.23 mm $°C^{-1}$ $d^{-1}$) to achieve realistic snow storage simulations.

Additionally, we provided a simple evaluation of the simulated snow water equivalent and snow depth (measured automatically during a single summer season) as described in L208-211 and Figure 3 (previously placed as Fig. A1 in the original version). Despite the lack of data, the modelled SWE values correlated well with measured ones.

To better address all these issues described above, we reformulated the methods section related to the HBV model description (L 153-162) and respective discussion (L 378-382). Among others, we have included the calibrated degree-day factor values.

Line 347: It should be …. was also slightly negative in 2014/15, ……………..

Thank you, we have corrected the sentence on line 401.

Line 435: Missing the reference of Seibert and Vis (2012).

We are not sure whether we interpret this comment correctly since the referenced line (L435 of the original manuscript) represents the beginning of the references section. The mentioned reference is included in the reference list (L694).

---

## Referee Report (RR1)

I commend the authors on a large amount of work, a well-written manuscript, and a thoughtful response to previous reviewers. I acknowledge how challenging it is to have a new reviewer come in mid-review and empathize with the authors on this. After reviewing the original and revised manuscripts as well as the response to reviewers. With that I've got some minor comments and some more major comments on figures.

Abstract/Introduction:

I find these sections very well-written. In general, I think a clearer statement of why we need to know about runoff processes in the Antarctic Peninsula would increase readability. Specifically, after the first sentence in the abstract and around line 30. I think there's a degree of the importance that's implicit, but I think clearly stating why runoff processes matter would help your reader. I'm more familiar with why runoff processes matter in populated/agricultural settings in terms of food production, water supply, infrastructure, and hazards (drought/floods) so having a clear statement would be helpful to reorient myself.

Methods:

I have concerns about the limitations in data availability for calibration, but recognize the limitations of field work and also commend the authors on a comprehensive response to previous questions. I have a few minor questions:

1. On line 153 is the single threshold temperature 0 deg C?
2. I'm still unclear about the role of PET as it appears to be estimated in the results, but I don't see any mention of how those data are estimated in methods. What are the key components of water vapor losses (i.e., is it sublimation given the relative aridity and temperature in the region)?

Results

1. Are you assuming that any KGE value > 0 is "good" consistent with Knoben (line 195)? https://hess.copernicus.org/articles/23/4323/2019/ If so, I'd state that as I think it's also somewhat common to assume that >0.5 is "good" for KGE.
2. Since you define the WY as beginning on June 1, I would suggest just adopting a single year to describe each WY consistent with common practice outside the Antarctic to simplify labeling on figures.
3. Most units are presented in mm, but SWE is sometimes in cm. Not clear if there's a reason for that, but flagging just in case.
4. I find all figures in the results to be very challenging to read/interpret given the overlap between daily and cumulative reporting and labeling. I know figures are such a challenge and that this was flagged in the previous review, but I think the readability of these figures is my largest issue with the manuscript.

   a. Figure 4: Cyan is a very challenging color to read and it took me a very long time to understand how to interpret the columns for panel a versus the line. There is so much going on in this figure that grasping the relationships between WE and SWE and T is really hard. Specific thoughts:

   i. at a minimum that there is greater space between panel a and b as I don't think that will disrupt the readers' ability to view relationships, but may make the figure less overwhelming.

   ii. Do you need to report annual WE since all other variables are daily or 10-day? I

   iii. t looks as though the figure has two grey bars on the right and left.

b. Figure 5: Again, there's simply too much packed into this figure for me to grasp the key points and the presentation of both daily and cumulative results is sort of challenging. Specific thoughts:

   i. Could you separate the cumulative curves out and construct a panel b presenting all cumulative curves, including P with T plotted on top as is?

   ii. I think the filled in area graphs are really tough to read since Q_snow and Q_glacier overlap so much at the beginning of the water year.

     1. I don't fully understand how the cumulative curve for Q_glacier is so close to Q_rain when the area plots seem to imply that it's closer to Q_snow unless they're stacked in which case I still find the plots confusing.

c. Figure 6: I appreciate the attempt to include the contributing factors to Q within the Q_total, but I find this to be a challenging figure to read as well. This may be the best way to present this figure and I'm sure it took the authors a tremendous amount of work to get here, but it's still really hard to read. Specific thoughts:

   i. think part of it relates to color and in many cases the lower bound of the error bars is very hard to determine.

   ii. I would recommend adding discussion of uncertainty to your discussion around lines 255 since based on my read of this figure, uncertainty around total Q for peaks is close to 2x the estimated Q in some cases (e.g., Dec).

   iii. Also explaining why uncertainties in Q_snow are so large relative to other components would be helpful.

d. Table 3: Can you explain bolded values in the caption. Also I don't know if this is an EGU formatting point, but usually I see table captions above tables.

e. Figure 7: Same comments about cumulative and daily values and area graphs as above. It seems like daily values are most important give the text? Could you create a third panel comparing cumulative curves to highlight the role of Q_glacier?

f. Figure 9: Why doesn't panel a have all the same boxes as panel b? Can you explain this in the text.

   i. I'm confused on Line 310 (the text implies three pairs, but the reporting is for only two? I'm not sure which pairing the reported stats correspond to. I would also say that it's slightly confusing to report a spearmans of 0.68 as 'strong,' but a spearmans of 0.63 as weaker. I would argue that both indicate moderate correlation with 0.82 indicating strong correlation.

g. Table 4: Could this be combined with previous tables/figures somehow. It's quite a lot of have 4 tables and 10 figures.

   i. I also don't follow line 321-323 completely. If Q_snow is the dominant driver of runoff (76% by my read) and Q_glacier drops as low as 1% per Section 3.4 couldn't it also be that smaller changes in Q_snow drive variability. That seems

like at least one plausible explanation. I don't see the direct line between inter-annual variability and runoff based on results and it seems like a big leap to me to state that that's the primary driver based on coefficients of variation when the proportional contributions of those components are so different.

    h. Figure 10: I think this figure is really interesting, but also challenging to read. I'd suggest renaming WY to a single year to reduce text here. I'm also not immediately sure what the quadrants are supposed to represent other than deviations from x=0, y=0? I think I understand that circles are meant to emphasize positive anomalies in T in recent years, but honestly I find it confusing. I'd rather have the color scale be consistent and the y-axis vary since it really takes a while to see how these points diverge. Also I think relative anomaly needs to be more clearly explained as it is in the text above Figure 10. This figure takes a lot of thinking to digest and I think anything you can do to help your reader get to the main point sooner would be very helpful

5. I sincerely appreciate the author's transparency in their discussion of limitations. It makes me trust your science and results to be so upfront. With that said, I do think that it distracts from your key findings to have it take up such a prominent role in your discussion (e.g., that's what I get to first, so I immediately am thinking about your limitations rather than what you're showing). Could you move this lower in your discussion (at the end)?

---

## Author Response (AR2)

**Authors' response to the Reviewer 3**

Black text: Reviewer's comment

Blue text: Authors' response

I commend the authors on a large amount of work, a well-written manuscript, and a thoughtful response to previous reviewers. I acknowledge how challenging it is to have a new reviewer come in mid-review and empathize with the authors on this. After reviewing the original and revised manuscripts as well as the response to reviewers. With that I've got some minor comments and some more major comments on figures.

Thank you for the review of our manuscript. We appreciate your constructive comments and suggestions. Please find our point-by-point response below. The line numbers in our responses are related to the revised version.

Abstract/Introduction:
I find these sections very well-written. In general, I think a clearer statement of why we need to know about runoff processes in the Antarctic Peninsula would increase readability. Specifically, after the first sentence in the abstract and around line 30. I think there's a degree of the importance that's implicit, but I think clearly stating why runoff processes matter would help your reader. I'm more familiar with why runoff processes matter in populated/agricultural settings in terms of food production, water supply, infrastructure, and hazards (drought/floods) so having a clear statement would be helpful to reorient myself.

We agree, and we added some more text both to the abstract (L11-12) and introduction (L40-43) sections highlighting the need to study hydrological processes in such remote areas, which, at the same time, belong to fragile environments that undergo major changes as a response to climate changes.

Methods:

I have concerns about the limitations in data availability for calibration, but recognize the limitations of field work and also commend the authors on a comprehensive response to previous questions. I have a few minor questions:

1. On line 153 is the single threshold temperature 0 deg C?

As stated on line 158 of the revised manuscript, "The TT is one of the calibrated model parameters." The median value resulting from 100 calibration runs was -0.21. We have added this information on line 466.

2. I'm still unclear about the role of PET as it appears to be estimated in the results, but I don't see any mention of how those data are estimated in methods. What are the key components of water vapor losses (i.e., is it sublimation given the relative aridity and temperature in the region)?

Thank you for the comment. The temperature-based method defined by Oudin et al. (2005) was used for the calculation of daily PET using observed air temperature measured at the Johann Gregor Mendel Station. The model then calculates actual evaporation from the input PET as a function of the simulated soil moisture. The snow routine does not account for either sublimation or evaporation from snow cover. We have reformulated the existing text or added a new text to the respective 2.3 Section (L151-152, 154 and 162-163).

Results

1. Are you assuming that any KGE value > 0 is "good" consistent with Knoben (line 195)? https://hess.copernicus.org/articles/23/4323/2019/ If so, I'd state that as I think it's also somewhat common to assume that >0.5 is "good" for KGE.

We agree and have reworded lines 201-202 accordingly.

2. Since you define the WY as beginning on June 1, I would suggest just adopting a single year to describe each WY consistent with common practice outside the Antarctic to simplify labeling on figures.

We are aware that an indication of WY by a single year would make the text and figure captions easier to read (actually, we have considered it already while writing the original manuscript). However, we have chosen the two-year indication mostly because it is consistent with other glaciological studies. Besides, a single-year naming might raise questions about which period is meant since 1$^{st}$ June as a starting date is quite close to the middle of the calendar year (opposite to the typical WY starting date e.g. in the northern hemisphere.) For these reasons, we prefer to keep two-year naming to avoid confusion, although it results in longer text and figure captions.

3. Most units are presented in mm, but SWE is sometimes in cm. Not clear if there's a reason for that, but flagging just in case.

We checked the text and found that the unit of cm is only used for snow depth. All SWE are always reported in mm.

4. I find all figures in the results to be very challenging to read/interpret given the overlap between daily and cumulative reporting and labeling. I know figures are such a challenge and that this was flagged in the previous review, but I think the readability of these figures is my largest issue with the manuscript.

We agree that some of our figures may require more time to look at, as they contain a lot of information. However, we feel that the information displayed in the figures is important to correctly interpret our results. Nevertheless, according to your suggestions, we reconsider the figures again and test several alternatives to improve the readability (see our detailed responses below).

a. Figure 4: Cyan is a very challenging color to read and it took me a very long time to understand how to interpret the columns for panel a versus the line. There is so much going on in this figure that grasping the relationships between WE and SWE and T is really hard. Specific thoughts:
i. at a minimum that there is greater space between panel a and b as I don't think that will disrupt the readers' ability to view relationships, but may make the figure less overwhelming.

We agree and have increased the space between the panels and changed the y-axis label in panel a) to make it clearer.

ii. Do you need to report annual WE since all other variables are daily or 10-day?

We believe that the annual change of WE is important because it represents essential information on glacier mass balance which is frequently used in other glaciological studies enabling the direct comparison of our results. In addition, the glaciological data used for model calibration were measured annually, so the change in annual WE also provides a direct reference to evaluate the model performance.

iii. It looks as though the figure has two grey bars on the right and left.

We were not able to find these bars in our original tiff figures. Perhaps, there might be some errors in the exported pdf of the original manuscript.

b. Figure 5: Again, there's simply too much packed into this figure for me to grasp the key points and the presentation of both daily and cumulative results is sort of challenging. Specific thoughts:
i. Could you separate the cumulative curves out and construct a panel b presenting all cumulative curves, including P with T plotted on top as is?

We tested your suggestion of creating the additional panel for the cumulative lines, but we found it less informative. In our opinion, the information from the daily values (areas) and the cumulative values (lines) complement each other well, so we prefer to keep them together.

ii. I think the filled in area graphs are really tough to read since Q_snow and Q_glacier overlap so much at the beginning of the water year.

The areas are stacked, so they do not overlap. We have added this important information to the figure caption. At the beginning of the water year, the runoff (if even occurs) is caused by snowmelt.

1. I don't fully understand how the cumulative curve for Q_glacier is so close to Q_rain when the area plots seem to imply that it's closer to Q_snow unless they're stacked in which case I still find the plots confusing.

As mentioned in our previous response, the areas are stacked (opposite to the lines). We have added this information to the figure caption to make it clearer.

c. Figure 6: I appreciate the attempt to include the contributing factors to Q within the Q_total, but I find this to be a challenging figure to read as well. This may be the best way to present this figure and I'm sure it took the authors a tremendous amount of work to get here, but it's still really hard to read. Specific thoughts:

i. think part of it relates to color and in many cases the lower bound of the error bars is very hard to determine.

We are aware, that the lower bounds of the whiskers may be difficult to see. This is because the related values are often small, even close to zero. In our opinion, it is not important to determine the exact value from the figure, as the overall variability of monthly runoff is evident. The colours were chosen to make them consistent with other figures (we tested different alternatives).

Nevertheless, we removed the horizontal lines from the figure which we hope makes the figure more readable.

ii. I would recommend adding discussion of uncertainty to your discussion around lines 255 since based on my read of this figure, uncertainty around total Q for peaks is close to 2x the estimated Q in some cases (e.g., Dec).

We agree, and we added some more text related to the monthly variability of runoff (lines 267-270).

iii. Also explaining why uncertainties in Q_snow are so large relative to other components would be helpful.

The whiskers in Fig. 6 do not represent uncertainty, but the inter-annual variability. Nevertheless, we agree that the variability in absolute numbers is largest for Qsnow. The possible explanation is provided in newly added lines 265-270.

d. Table 3: Can you explain bolded values in the caption. Also I don't know if this is an EGU formatting point, but usually I see table captions above tables.

The values in bold are the minimum and maximum values. We have added this information to the table caption. We did not find a specific rule regarding table captions placement for TC. However, we moved the caption above the table according to common praxis.

e. Figure 7: Same comments about cumulative and daily values and area graphs as above. It seems like daily values are most important give the text? Could you create a third panel comparing cumulative curves to highlight the role of Q_glacier?

We tested different alternatives but found the existing one as most informative. We also refer to our responses regarding Figure 5.

f. Figure 9: Why doesn't panel a have all the same boxes as panel b? Can you explain this in the text.

Please note that panel a) shows correlations of the annual values of the selected variables, while b) shows correlations for daily values of the selected variables. Therefore, the sum of positive temperatures (Tpositive) cannot be included in panel b) because, by definition, it represents a seasonal variable. As for global radiation (GR), the data were available only for the period from 1 January 2015 to 10 March 2018 thus there is not enough data to investigate annual correlations with other variables available for a longer period (see line 332). We have included this latter information in the figure caption. Additionally, we somewhat reworded the text related in lines 319-325 to provide more context (reflecting also your comment below).

i. I'm confused on Line 310 (the text implies three pairs, but the reporting is for only two? I'm not sure which pairing the reported stats correspond to. I would also say that it's slightly confusing to report a spearmans of 0.68 as 'strong,' but a spearmans of 0.63 as weaker. I would argue that both indicate moderate correlation with 0.82 indicating strong correlation.

Thank you, you are right there was a mistake in the text. Additionally, we have reformulated the whole text in lines 319-325 for clarity.

g. Table 4: Could this be combined with previous tables/figures somehow. It's quite a lot of have 4 tables and 10 figures.

We agree. We deleted the table and added its content directly to the text instead (lines 347-350).

i. I also don't follow line 321-323 completely. If Q_snow is the dominant driver of runoff (76% by my read) and Q_glacier drops as low as 1% per Section 3.4 couldn't it also be that smaller changes in Q_snow drive variability. That seems like at least one plausible explanation. I don't see the direct line between inter-annual variability and runoff based on results and it seems like a big leap to me to state that that's the primary driver based on coefficients of variation when the proportional contributions of those components are so different.

We have changed Fig. 10 and related interpretation in abstract (lines 20-22) results (lines 340-350), discussion (lines 413-418) and conclusions (line 485). Please see our response to your next comment for details.

h. Figure 10: I think this figure is really interesting, but also challenging to read. I'd suggest renaming WY to a single year to reduce text here. I'm also not immediately sure what the quadrants are supposed to represent other than deviations from x=0, y=0? I think I understand that circles are meant to emphasize positive anomalies in T in recent years, but honestly I find it confusing. I'd rather have the color scale be consistent and the y-axis vary since it really takes a while to see how these points diverge. Also I think relative anomaly needs to be more clearly explained as it is in the text above Figure 10. This figure takes a lot of thinking to digest and I think anything you can do to help your reader get to the main point sooner would be very helpful

Thank you for this detailed comment. Regarding the WY renaming we refer to our respective response above.

We agree that the original Fig. 10 was not fully clear since it mixed different types of information in individual panels (relative Q anomaly in panel (a) compared to relative anomalies of Q fractions in

other panels, which, by definition, already represent relative values). Therefore, we decided to show anomalies of runoff components in panels b), c), and d) to be consistent with panel a). Additionally, we have substantially rewritten the related text on lines 340-350 (results section) and 413-418 (discussion section).

5. I sincerely appreciate the author's transparency in their discussion of limitations. It makes me trust your science and results to be so upfront. With that said, I do think that it distracts from your key findings to have it take up such a prominent role in your discussion (e.g., that's what I get to first, so I immediately am thinking about your limitations rather than what you're showing). Could you move this lower in your discussion (at the end)?

We like your suggestion, thank you. We moved the data and modelling uncertainty subsection to the end of the discussion section.

---

## Author Response (AR3)

Thank you very much for your thoughtful review of many of my queries about your manuscript. I am, on the whole, satisfied with these responses. I do have a few points that remained unclear in my rereading of the manuscript that I hope are helpful in extending the reach of this work.

Thank you for your comments and suggestions. Please find our point-by-point response below (in blue). The line numbers in our responses are related to the revised version.

Q_glacier versus Q_snow conclusion:
1. Q_glacier to Q variability: Could you add something discussion about the limitations here. Mostly I'm interested in why the correlation between daily runoff and daily Q_glacier is much weaker than other runoff components, but the annual associations seem to drive a key finding of the paper. This seems particularly critical given that there are overall very few points and the majority of them are modeled based on limited observational data (which you do a nice job of explaining!)

a. As someone predominately focused on snow versus glaciers, I'm genuinely curious about whether we would expect associations at different timescales. Moreover, what does that mean for conclusions about inter-annual variability (if anything)?

Thank you for the comment. It is difficult to fully identify the drivers leading to different runoff responses on different time scales as more detailed in-situ observations of the runoff components are missing in the study area.

Nevertheless, one of the reasons for the lower correlations between daily runoff and daily $Q_{glacier}$ compared to other runoff components is probably that glacier melt in each elevation zone is simulated only in the case of the snow-free glacier (as mentioned on L166). In contrast to snow and rain contributions, this resulted in a reduced number of days in the study period where $Q_{glacier}$ contributed to total runoff (57% of days with non-zero runoff) leading to a reduced correlation coefficient.

Additionally, glacier runoff often follows after snowmelt runoff, so total runoff on a given day often includes both glacier and snowmelt runoffs from that day (snowmelt runoff occurs from higher elevations zones where the snow is still present) and snowmelt runoff from previous days (the response time is controlled by the calibrated MAXBAS parameter of the HBV model response function). All these factors cause the total runoff during glacier melt to be mixed from different water sources, resulting in a lower correlation coefficient.

On seasonal and annual scales, $Q_{glacier}$ is driven by air temperature and has a similar variability as total runoff. In warm years, snow melts first, followed by glacier ice, leading to higher total runoff. In contrast, in colder years, snow melt still occurs (although slower and later than in warm years), but glacier melt is negligible or even absent, reducing its contribution to total runoff (as mentioned in L417). Thus, despite weaker daily correlations, there are strong annual associations of Q and $Q_{glacier}$.

We enlarged the related discussion section in the revised manuscript to include the above explanations (L425-432, L496-498).

In addition to the above explanation, there are certainly some methodological aspects which might affect the correct interpretation of the results. Most of all we analyzed only an 11-year study period, which may not include a full range of different climate conditions. For example, warm and wet years are not present in this study period as shown in Fig. 10. We addressed this aspect better in L419-422.

2. Correlations between Q_snow and Q_rain and Q (Line 343). What do you mean "no specific pattern was found for Q_snow and Q_rain"? I find it slightly hard to compare these given that the color bar is the same, but the values represented by that color change.

a. For example, a dark blue in Figure 10c isn't a dark blue in 10b or d. I recognize that they would all be blue, but I find the saturation and coloring to overemphasize the correlations a bit since by my interpretation that would diminish the blues in Quadrant 4. I'm not trying to nitpick it just seems important given that there are so few points and the axes aren't consistent.

The phrase "no specific pattern…" was a relic of the previous version of the manuscript which we forgot to rephrase, thank you for double checking. We have reformulated it on L347-348.

Regarding Fig. 10., we have prepared several alternatives to this figure (some of them already for the previous round of reviews related to one of your comments). After careful consideration of the pros and cons, we think the one we used in the previous revised manuscript is the most informative. We are aware that using the same color map for different value ranges could potentially lead to misleading interpretations. However, we believe that the color bar labels and the disclaimer in the figure caption should be sufficient for the reader to interpret the figure correctly. Therefore, we consider the current version of the figure to be an acceptable compromise. Individual alternatives are shown at the end of this response together with our comments.

Figures:
I understand and accept that the presentation of cumulative and daily results is the best format according to your investigations, however, I would strongly consider a different color palette—I still find these very challenging to read and I'm likely spending much more time on them than your average reader (e.g., Figure 5 and 7).

We understand that it might be difficult to present as much information as possible in one plot by preserving readability (and keeping the color pallet of the individual variables the same across all figures in the manuscript, if possible). Besides, we need to keep the colors to be readable for different color blindnesses (as required by the journal). Therefore, it is not easy to meet all these expectations. Nevertheless, we have changed the colors and styling of the figures and we hope that they are more readable now. Regarding the above changes, we also modified Figures 4, 6 and 8 so that each variable has the same color in all figures.

1. For Figure 7, it's not clear to me that there's immediate value in having the axes and text in different colors, particularly if your reader may have challenges with different color palettes.

The colored axis has been implemented based on Reviewer 1's suggestion as it made sense to us. There are also labels to make it clear what each axis refers to. We checked this figure with the color blindness simulator and did not find any potential problems. Therefore, we prefer to keep the figure as is.

2. In the case of Figure 10 (per the above), I'm curious as to whether you considered my suggestion of a different color palette for the different scales or at least having the color associated with the same value across scales as I find it slightly misleading that dark blue can be both 20% and 100%.

As mentioned in our response above, we tested several options. We created two versions of the figure with the same color scale for all panels (Fig. 1 and Fig. 2 of this response). Fig. 1 of this response has a value range from -40 to 40. Although it shows the above (below) average years well, we found this figure misleading because it seems that all variables have the same variance.

[Figure]

Fig 1: Manuscript Fig. 10 with the same color scale for all panels with the range (-40, 40).

In contrast, Figure 2 of this response has a scale of -160 to 160 covering the variance of the most variable component, so the colors are the same for all panels. While this is a good solution for b) and d) panels, the variability in a) and c) panels is poorly visible.

[Figure]

Fig. 2: Manuscript Fig. 10 with the same color scale for all panels with maximal range calculated from all data.

The third option (Fig. 3 of this response) represents the reviewer's suggestion (if we understood correctly). After consideration, we found the use of the two different color scales rather confusing as it suggests two different variables to be displayed in the figures.

[Figure]

Fig. 3: Manuscript Fig. 10 with two different color scales.

a. I would also note that it seems like T_positive is defined in different ways on lines 191, 341, and 416. It took me a little bit to work out that I think it's the sum of positive temperatures over an annual period (I think*).

We agree with the reviewer. $T_{positive}$ is the annual sum of positive daily temperatures as it is defined in L191. We have rewritten the sentence on L419-420 which we found confusing.